# Determining accurate conformational ensembles of intrinsically disordered proteins at atomic resolution

Kaushik Borthakur [1], Thomas R. Sisk[1], Francesco P. Panei[2], Massimiliano Bonomi [2] & Paul Robustelli [1] ✉

Determining accurate atomic resolution conformational ensembles of intrinsically disordered proteins (IDPs) is extremely challenging. Molecular dynamics (MD) simulations provide atomistic conformational ensembles of IDPs, but their accuracy is highly dependent on the quality of physical models, or force fields, used. Here, we demonstrate how to determine accurate atomic resolution conformational ensembles of IDPs by integrating all-atom MD simulations with experimental data from nuclear magnetic resonance (NMR) spectroscopy and small-angle x-ray scattering (SAXS) with a simple, robust and fully automated maximum entropy reweighting procedure. We demonstrate that in favorable cases, where IDP ensembles obtained from different MD force fields are in reasonable initial agreement with experimental data, reweighted ensembles obtained with this approach converge to highly similar conformational distributions. The maximum entropy reweighting procedure presented here facilitates the integration of MD simulations with extensive experimental datasets and demonstrates progress towards the calculation of accurate, force-field independent conformational ensembles of IDPs at atomic resolution.

Many proteins that perform important biological functions are completely or partially disordered under physiological conditions[1,2]. These so-called intrinsically disordered proteins (IDPs) lack a well-defined tertiary structure in solution and instead populate a conformational ensemble of rapidly interconverting structures. Structurally characterizing the heterogeneous conformational ensembles adopted by IDPs can provide mechanistic insight into their physiological interactions and functions[3,4]. IDPs are implicated in many human diseases and are increasingly being pursued as drug targets[5]. Determining accurate conformational ensembles of IDPs and IDPs in complex with small molecules can provide valuable insight for the rational design of IDP inhibitors[6–11].

Experimentally determining atomic-resolution conformational ensembles of IDPs is extremely challenging. Most experimental techniques used to structurally characterize IDPs in solution, such as

nuclear magnetic resonance (NMR) spectroscopy and small-angle X-ray scattering (SAXS), report on conformational properties averaged over many molecules over long periods of time[12]. Such *ensemble-averaged* measurements can be consistent with a large number of conformational distributions. Typical experimental datasets used to characterize conformational ensembles of IDPs are also sparse, meaning that they report on a small subset of structural properties of IDPs. Many experimental data used to characterize IDPs, such as NMR chemical shifts, are challenging to interpret and predict as they are sensitive to a combination of many structural properties[13–17].

Molecular dynamics (MD) computer simulations are a powerful approach for determining atomic-resolution conformational ensembles of IDPs in silico. In principle, long timescale or enhanced sampling all-atom MD simulations of an IDP run with an accurate physical model, or *force field*, can provide atomically detailed structural descriptions of

[1]Department of Chemistry, Dartmouth College, Hanover, NH, USA. [2]Institut Pasteur, Université Paris Cité, CNRS UMR 3528, Computational Structural Biology Unit, Paris, France. ✉e-mail: paul.j.robustelli@dartmouth.edu

the conformational states populated in solution, along with their equilibrium populations. In practice, MD simulations are limited by the accuracy of the force fields used to describe the interactions between atoms in molecules. Recent improvements in molecular mechanics force fields and water models have dramatically improved the accuracy of MD simulations of IDPs as assessed by agreement with a large variety of experimental measurements[18–23]. However, discrepancies between simulations and experiments remain among the best performing force fields[18–23].

Due to the challenges of determining conformational ensembles of IDPs from experimental or computational methods alone, integrative approaches, where experimental data are used to construct or refine computational models of IDP ensembles, have grown increasingly popular[24–36]. The maximum entropy principle[37] is the basis for a number of successful *reweighting*[24–32,38] and *biasing*[39–43] approaches to determine conformational ensembles of proteins. In the maximum entropy framework, one seeks to introduce the minimal perturbation to a computational model required to match a set of experimental data. Integrative methods based on the maximum entropy principle have provided valuable insight into structural ensembles of IDPs[6,10,38,42–45]. Several methodological challenges have, however, hindered the development of fully general, prescriptive, and automated maximum entropy methods for calculating atomic-resolution conformational ensembles of IDPs from an arbitrary set of experimental data. In particular, when integrating experimental data from several sources, researchers are often required to make subjective decisions about the importance of satisfying different types of restraints that can strongly influence the structural properties of the calculated ensembles.

These methodological challenges have obscured an important question in IDP structural biology: with sufficient experimental data, can we determine physically realistic atomic-resolution IDP ensembles with conformational properties that are independent of the force fields used to generate the initial computational models? This question is particularly timely, as there is a growing interest in leveraging AI and deep generative models to predict conformational ensembles of flexible biomolecules[46]. Molecular simulations are increasingly being used to train deep generative models to predict the conformational ensembles of IDPs[47,48]. Presently, there is little consensus on what constitutes a "ground truth" conformational ensemble of an IDP, and these methods have largely been trained solely on computational models. If it becomes feasible to determine accurate, force-field independent IDP ensembles at atomic resolution, these ensembles could provide valuable training and validation data for machine learning methods to predict atomic-resolution conformational ensembles of IDPs, facilitating the development of efficient alternatives to MD for generating conformational ensembles of IDPs in the same way that methods like AlphaFold3[49] provide accurate structural models of folded proteins.

Here, we introduce a simple, robust, and fully automated maximum entropy reweighting procedure to determine accurate atomic-resolution conformational ensembles of IDPs by reweighting MD simulations with extensive experimental datasets from NMR and SAXS. The proposed method effectively combines restraints from an arbitrary number of experimental datasets using a single adjustable parameter: the desired number of conformations in the calculated ensemble. We demonstrate that the proposed approach, which does not require manually tuning the strength of experimental restraints, produces atomic-resolution structural ensembles of IDPs with exceptional agreement with extensive experimental datasets. In addition to providing an effective approach to combine experimental data from multiple sources, the proposed method produces statistically robust IDP ensembles with excellent sampling of the most populated conformational states observed in unbiased MD simulations and minimal overfitting to experimental data.

We apply the proposed method to assess whether conformational ensembles derived from long-timescale MD simulations run with different state-of-the-art force fields converge to similar conformational distributions upon reweighting with extensive experimental datasets. We determine conformational ensembles of five well-studied IDPs by reweighting unbiased long-timescale MD simulations run with different state-of-the-art force fields[19] and introduce an approach to quantify the similarity of IDP ensembles. For three of the five IDPs studied, we find that the calculated conformational ensembles are highly similar and can be considered a force-field independent approximation of the true underlying solution ensembles. In two of the IDPs studied, unbiased MD simulations performed with different force fields sample relatively distinct regions of conformational space, and our proposed reweighting method clearly identifies one ensemble as the most accurate representation of the true solution ensemble. This demonstrates that in favorable cases, where IDP ensembles obtained from different MD force fields are in reasonable initial agreement with experimental data, reweighted ensembles obtained with the proposed approach converge to highly similar conformational distributions.

In this work, we have proposed and validated a robust maximum entropy reweighting protocol to determine accurate atomic-resolution conformational ensembles of IDPs from unbiased MD simulations. We provide an in-depth comparison of IDP ensembles derived from different force fields and show that in favorable cases, IDP ensembles converge to highly similar conformational distributions after reweighting. We believe that these results represent substantial progress in IDP ensemble modeling and suggest that the field may be maturing from assessing the accuracy of disparate computational models towards the realm of atomic-resolution integrative structural biology. We anticipate that the reweighting protocol proposed here will provide a valuable tool for integrating MD simulations with extensive experimental datasets to determine accurate atomic-resolution conformational ensembles of IDPs and facilitate the calculation of force-field independent IDP ensembles.

## Results

### Determining accurate atomic-resolution ensembles of IDPs

We propose a simple, robust, and automated maximum entropy reweighting procedure to determine accurate atomic-resolution conformational ensembles of IDPs by integrating all-atom MD simulations with extensive experimental datasets from NMR and SAXS (see "Maximum entropy reweighting with a single free parameter" section). The rationale behind this approach is explained in detail in "Developing a maximum entropy reweighting protocol with a single free parameter" in the Supporting Information. A key component of the proposed approach is that the strengths of restraints from different experimental datasets are automatically balanced based on the desired number of conformations, or *effective ensemble size*, of the final calculated ensemble. The effective ensemble size is defined according to the *Kish ratio* ($K$) (Eq. 2), which is a measure of the fraction of conformations in an ensemble with statistical weights substantially larger than zero[27,50] (see "Kish ratio" section).

We use the proposed reweighting method to determine conformational ensembles of five IDPs that were previously used to benchmark the accuracy of recently developed force fields[19]: A$\beta$40[51] (40 residues), drkN SH3[52] (59 residues), ACTR[53] (69 residues), The ParE2-associated antitoxin (PaaA2)[54] (70 residues), and α-synuclein[55] (140 residues). These IDPs span a range of secondary structure propensity: A$\beta$40 and α-synuclein contain little-to-no experimentally detectable residual secondary structure, ACTR and drkN SH3 contain regions of residual helical structure, and PaaA2 contains two stable helical elements connected by a flexible linker. The experimental data used for reweighting and the methods used to calculate experimental data from IDP ensembles are described in "Experimental data used for

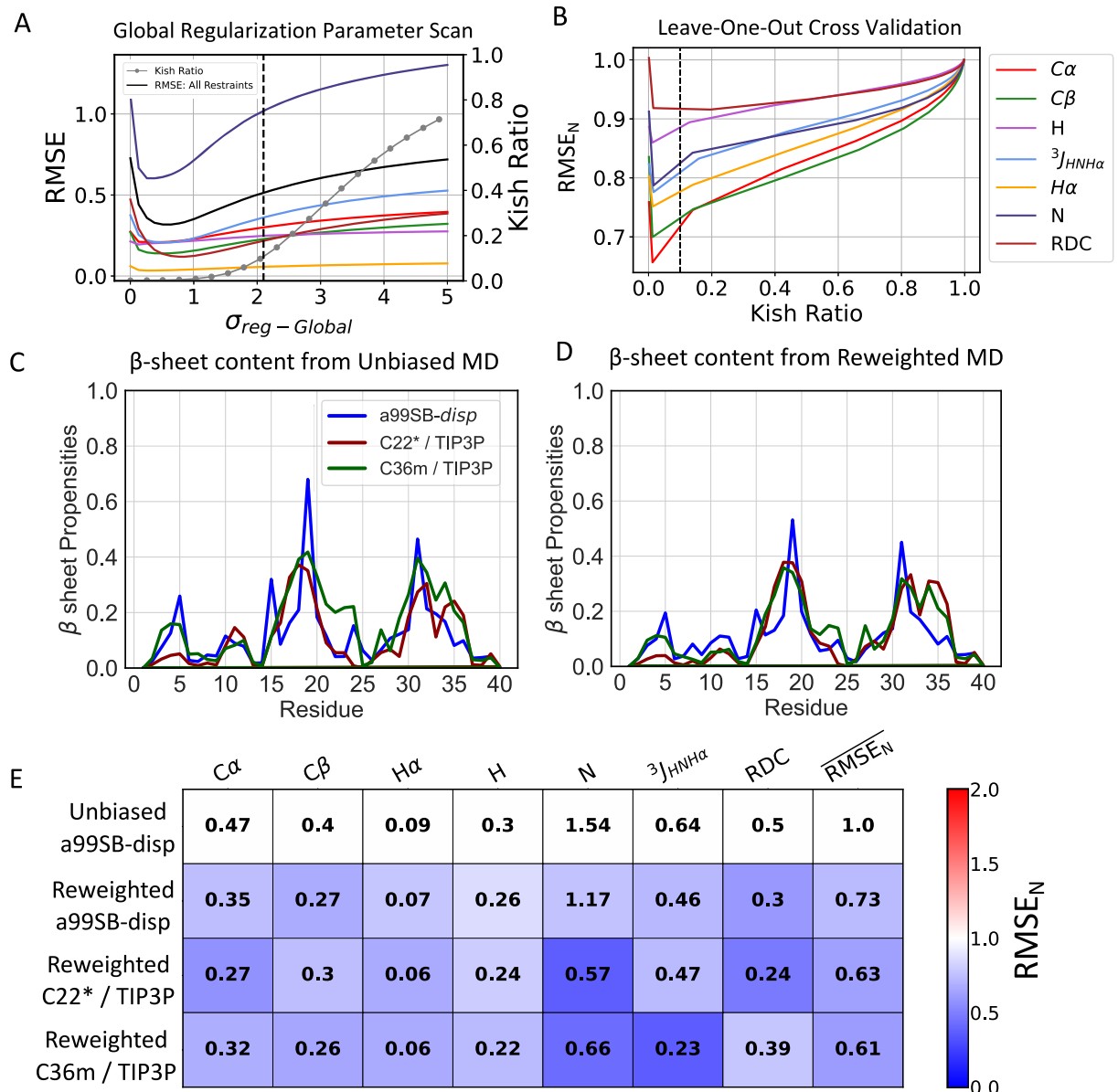

**Fig. 1 | Comparison of unbiased and reweighted MD ensembles of Aβ40. A** Root mean square error (RMSE) between calculated and experimental data in reweighted Aβ40 a99SB-*disp* MD ensemble as a function of the value of the global regularization parameter scaling factor ($\sigma_{reg-Global}$). The Kish ratio of each reweighted ensemble is indicated by gray dots. **B** The normalized RMSE (RMSE$_N$) of unrestrained data in leave-one-out cross-validation tests, where all other experimental data types are used as restraints for reweighting. RMSE$_N$ values of each data type are normalized by the RMSE observed in the unbiased a99SB-*disp* Aβ40 MD ensemble. Black dotted lines indicate the Kish ratio threshold $K = 0.10$ in each plot. Populations of β-sheets in unbiased MD ensembles and reweighted MD ensembles of Aβ40 are shown in (**C**) and (**D**), respectively. **E** Comparison of the RMSE between calculated and experimental data in reweighted Aβ40 ensembles derived from different force fields. Reweighted ensembles were calculated using all experimental data as restraints with a Kish ratio threshold of $K = 0.10$. Each square is colored to reflect the value of the normalized RMSE (RMSE$_N$) relative to the unbiased a99SB-*disp* MD ensemble.

reweighting" and "Calculating experimental observables" in the Supporting Information.

We determine conformational ensembles of Aβ40, drkN SH3, ACTR, PaaA2, and α-synuclein by reweighting 30μs MD simulations run with three different protein force fields and water model combinations: a99SB-*disp*[19] with a99SB-*disp* water, Charmm22*[56] with TIP3P water[57], and Charmm36m[22] with TIP3P water. We subsequently refer to these protein force field and water model combinations as a99SB-*disp*, C22*, and C36m, respectively. All unbiased MD ensembles contained 29,976 structures. We performed reweighting using a Kish Ratio threshold of $K = 0.10$, such that each reweighted ensemble contains ~3000 structures. Reweighted ensembles of each protein derived from a99SB-*disp*, C22*, and C36m force fields have been deposited in the protein ensemble database[58] (see "Data availability" section for accession codes). All experimental data used for reweighting and code used to perform reweighting and analyze reweighted ensembles are freely available from https://github.com/paulrobustelli/Borthakur_MaxEnt_IDPs_2024/.

We illustrate the key components of the proposed reweighting procedure in reweighting calculations performed on the a99SB-*disp* MD ensemble of Aβ40 (Fig. 1, Supplementary Fig. 3). We first use forward models[15,59–62] to predict the values of the $i$ experimental measurements as restraints in each frame of the unbiased MD ensemble. We calculate $\sigma_{i,MD}$, which reflects the statistical error of forward models' predictions from the unbiased MD ensemble, of each data point using the Flyvbjerg block analysis method[63]. We then

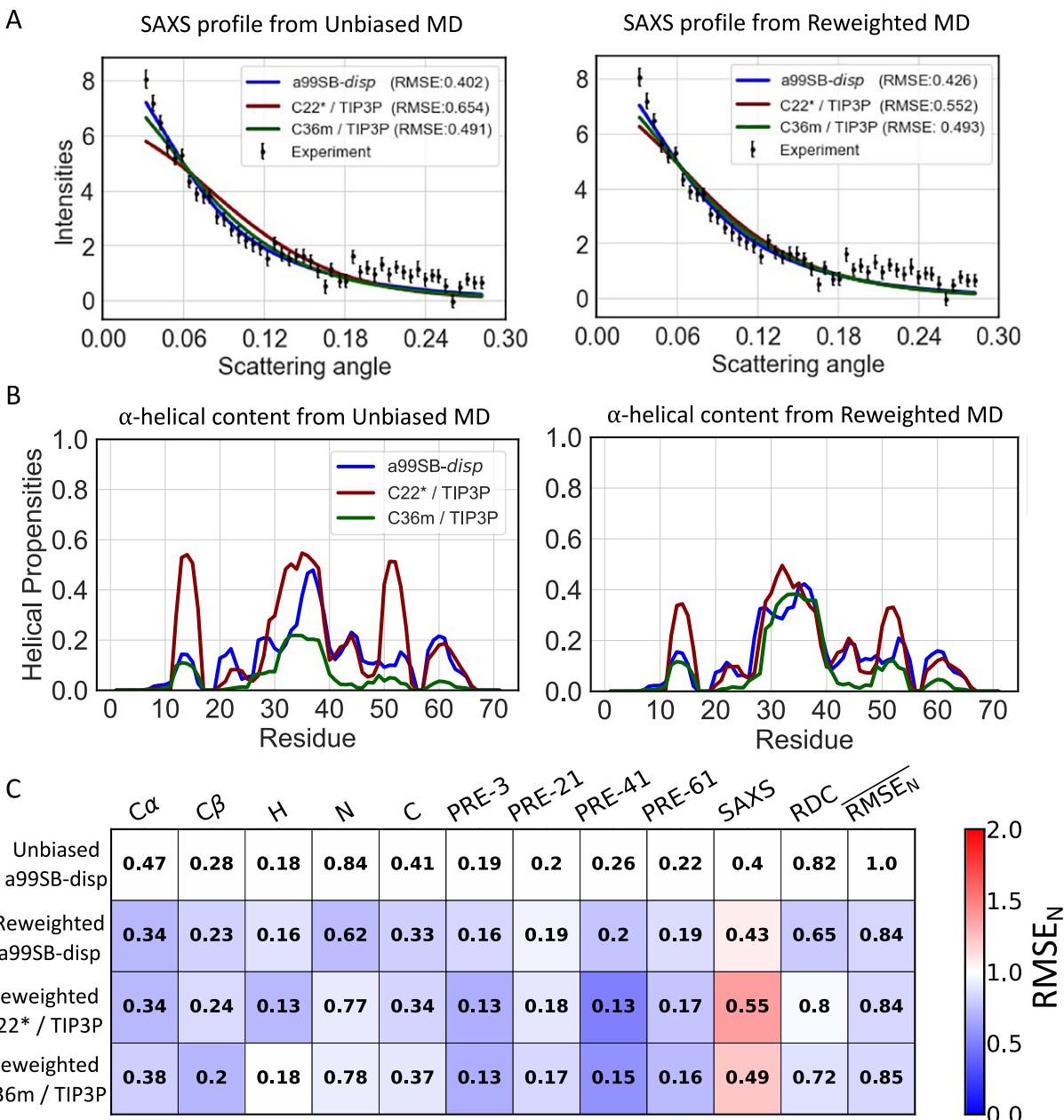

**Fig. 2 | Comparison of unbiased and reweighted MD ensembles of ACTR.**
**A** Comparison of SAXS profiles calculated from unbiased MD ensembles and reweighted MD ensembles of ACTR. Experimental SAXS profile of ACTR is displayed with error bars as reported in the original publication[74]. **B** Populations of α-helical conformations in unbiased MD ensembles and reweighted MD ensembles of ACTR. **C** Comparison of the RMSE between calculated and experimental data in reweighted ACTR ensembles derived from different force fields. Reweighted ensembles were calculated using all experimental data as restraints with a Kish ratio threshold of $K = 0.10$. Each square is colored to reflect the value of $RMSE_N$ relative to the unbiased a99SB-*disp* MD ensemble.

perform reweighting using each experimental data type (i.e., Cα chemical shifts, $J_{HNH\alpha}$, SAXS, etc.) as the only restraints and identify a value of the regularization parameter $\sigma_{reg}$ (Eq. 1) for each data type (i.e., $\sigma_{reg-C\alpha}$, $\sigma_{reg-J_{HNH\alpha}}$, $\sigma_{reg-SAXS}$, etc.) that produces a reweighted ensemble with a Kish ratio $K \geq 0.10$ (Supplementary Fig. 3). We subsequently perform reweighting using all experimental data as restraints, rescaling the regularization parameters of each data type by a global scaling factor $\sigma_{reg-Global}$, and identify the minimum value of $\sigma_{reg-Global}$ that produces a final reweighted ensemble with $K \geq 0.10$ (Fig. 1A).

The root mean squared error (RMSE) between calculated and experimental data decreases for all data types as we increase the strength of experimental restraints by decreasing the value of $\sigma_{reg-Global}$. This demonstrates that the proposed reweighting method simultaneously improves agreement with all experimental data types

in this system without requiring manual adjustment of the strengths of experimental restraints. We perform leave-one-out cross-validation tests for each experimental data type, and find that agreement with all withheld data improves upon reweighting, demonstrating that the proposed reweighting procedure results in minimal overfitting (Fig. 1B, Supplementary Fig. 3). We compare the accuracy of reweighted ensembles to unbiased MD ensembles using a normalized RMSE ($RMSE_N$) metric for each experimental data type and we define a global quality index, the averaged normalized RMSE ($\overline{RMSE_N}$), to compare the accuracy of the reweighted ensembles obtained from different force fields (see "Normalized RMSE ($RMSE_N$) comparisons" section). Further discussion of cross-validation results is included in the Supporting Information section "Developing a maximum entropy reweighting protocol with a single free parameter".

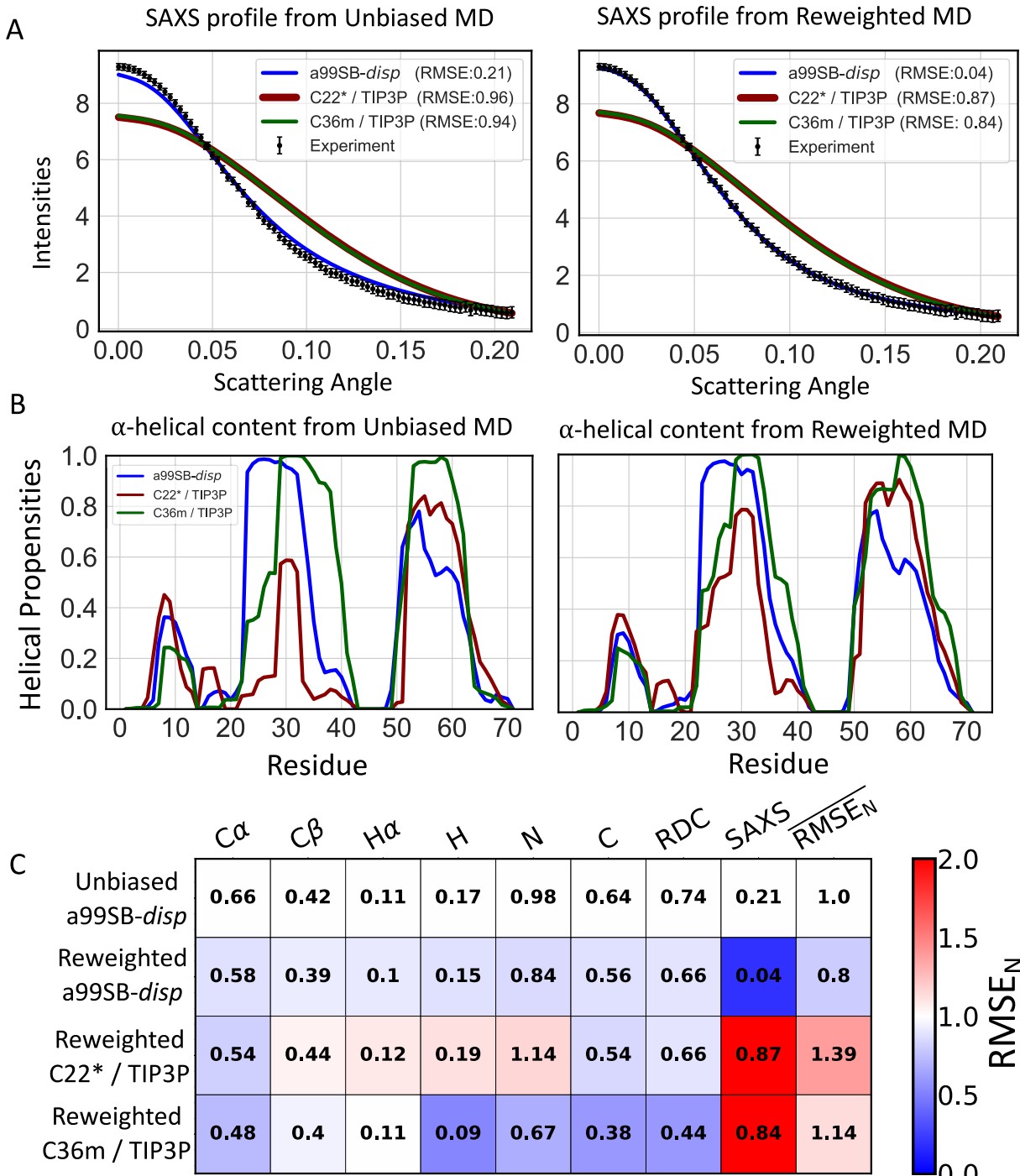

**Fig. 3 | Comparison of unbiased and reweighted MD ensembles of PaaA2.**
**A** Comparison of SAXS profiles calculated from unbiased MD ensembles and reweighted MD ensembles of PaaA2.Experimental SAXS profile of PaaA2 is displayed with error bars as reported in the original publication[54]. **B** Populations of α-helical conformations in unbiased MD ensembles and reweighted MD ensembles of PaaA2. **C** Comparison of the RMSE between calculated and experimental data in reweighted PaaA2 ensembles derived from different force fields. Reweighted ensembles were calculated using all experimental data as restraints with a Kish ratio threshold of $K = 0.10$. Each square is colored to reflect the value of $RMSE_N$ relative to the unbiased a99SB-*disp* MD ensemble.

We compare the accuracy of reweighted conformational ensembles of Aβ40, ACTR, PaaA2, drkN SH3, and α-synuclein derived from 30 μs unbiased a99SB-*disp*, C22*, and C36m MD simulations in Figs. 1, 2, 3, Supplementary Figs. 12 and 21, respectively. We find that when we apply our proposed reweighting method to IDP ensembles obtained from long-timescale MD simulations with reasonably accurate force fields, the reweighted conformational ensembles are in excellent agreement with extensive experimental NMR and SAXS datasets that describe both local and global structures of IDPs. We observe that for each protein studied here, we obtain at least one reweighted ensemble that is in substantially better agreement with experimental data than the most accurate unbiased MD ensembles of these proteins reported in previous benchmark studies[19,20]. Further comparisons of the accuracy of reweighted ensembles derived from different force fields and detailed comparisons of cross-validation metrics are included in

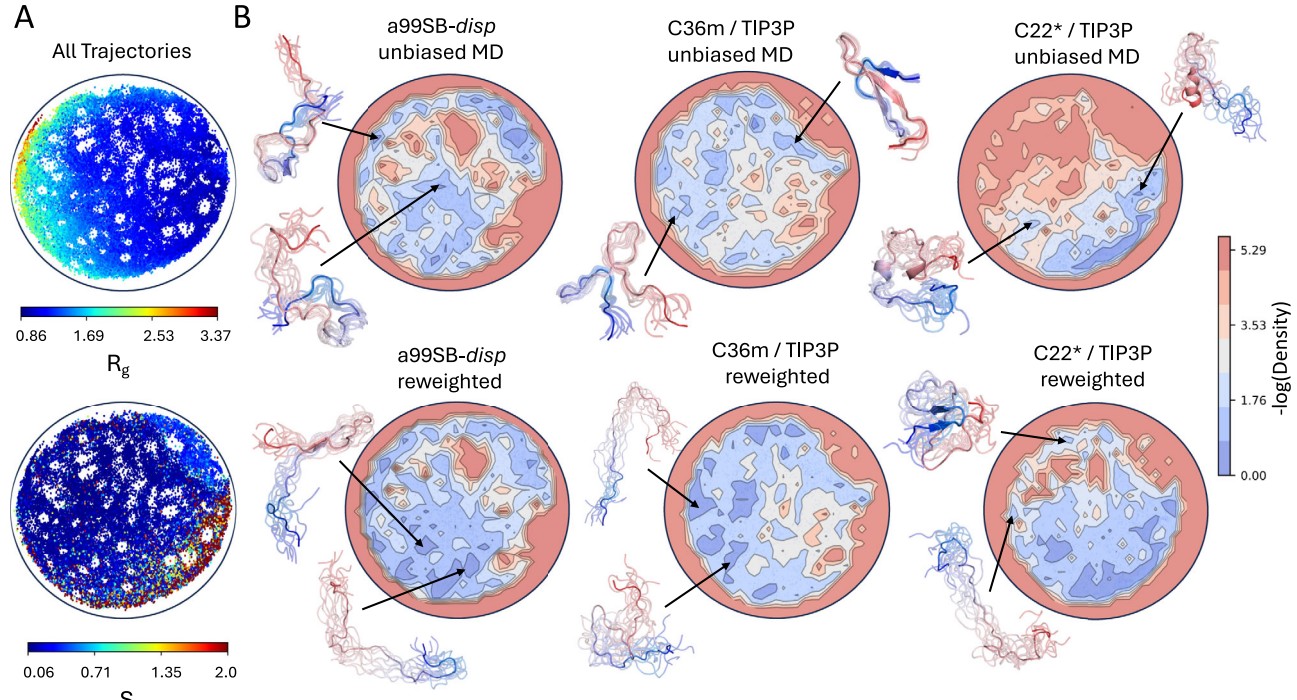

**Fig. 4 | Comparison of energy landscape visualization method (ELViM) projections of unbiased and reweighted MD ensembles of Aβ40.** A 2D ELViM embedding was calculated using all conformations from unbiased a99SB-*disp*, C22*, and C36m MD ensembles of Aβ40. **A** The values of the radius of gyration ($R_g$) and the α-helical order parameter $S\alpha$ of each conformation in the unbiased MD ensembles of Aβ40 projected onto the ELViM latent space. **B** Comparison of the density of ELViM projections of unbiased and reweighted MD ensembles of Aβ40. Representative structures are shown for regions of high density in ELViM projections of each ensemble. Structures are colored with a blue-to-red gradient from the N-terminus to the C-terminus.

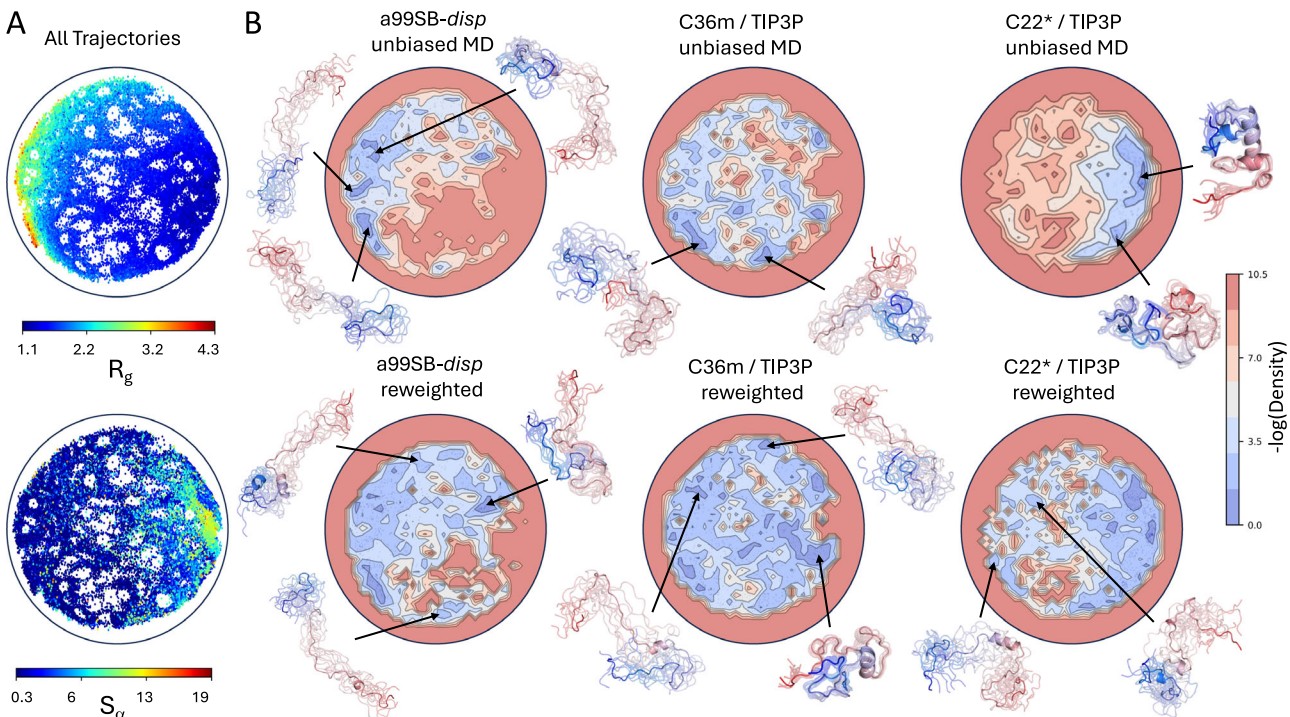

**Fig. 5 | Comparison of energy landscape visualization method (ELViM) projections of unbiased and reweighted MD ensembles of ACTR.** A 2D ELViM embedding was calculated using all conformations from unbiased a99SB-*disp*, C22*, and C36m MD ensembles of ACTR. **A** The values of the radius of gyration ($R_g$) and the α-helical order parameter $S\alpha$ of each conformation in the unbiased MD ensembles of ACTR projected onto the ELViM latent space. **B** Comparison of the density of ELViM projections of unbiased and reweighted MD ensembles of ACTR. Representative structures are shown for regions of high density in ELViM projections of each ensemble. Structures are colored with a blue-to-red gradient from the N-terminus to the C-terminus.

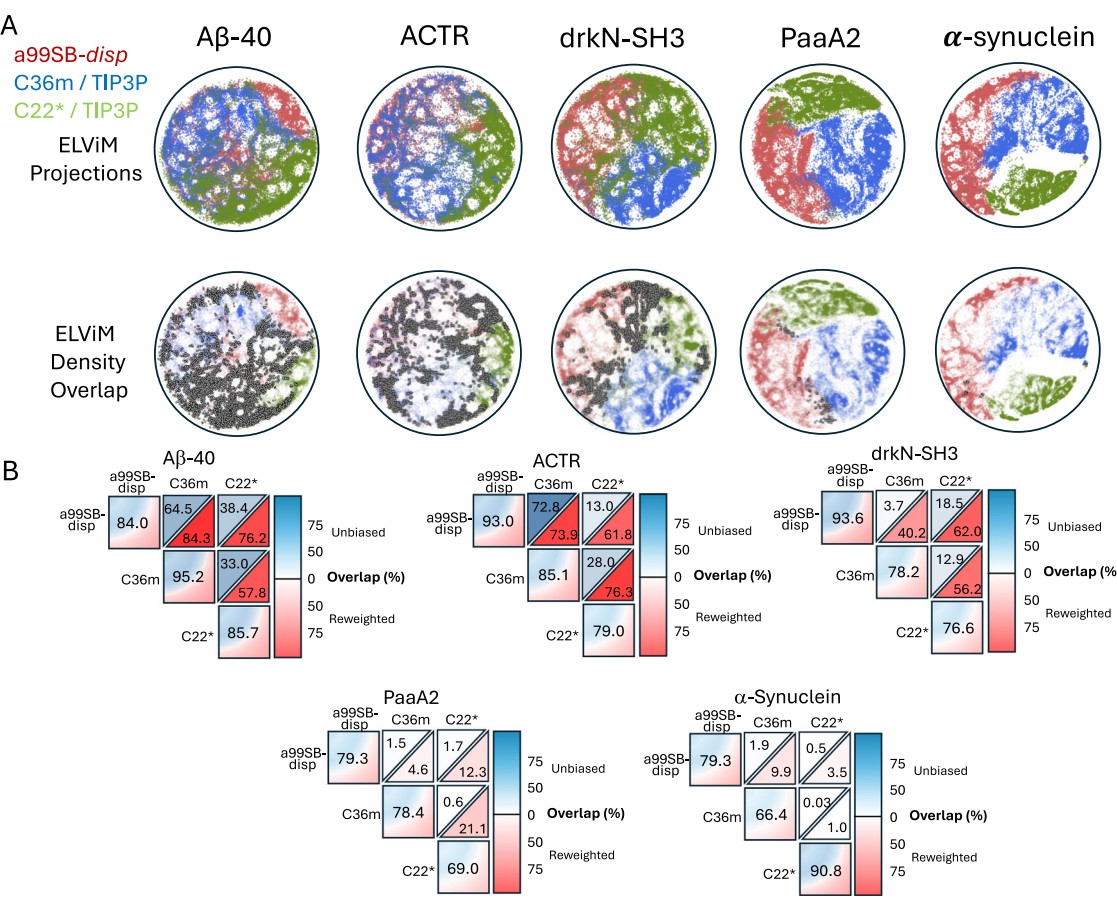

**Fig. 6 | Comparison of the overlap of energy landscape visualization method (ELViM) projections of unbiased and reweighted MD ensembles. A** 2D ELViM embeddings were calculated for Aβ40, drkN SH3, ACTR, PaaA2, and α-synuclein. ELViM embeddings were calculated for each protein using all conformations from unbiased a99SB-*disp*, C22*, and C36m MD ensembles. Projections of unbiased a99SB-*disp*, C22*, and C36m MD ensembles on ELViM latent space are shown in the top row. In the bottom row, each point is colored with an opacity proportional to its statistical weight after reweighting with all available experimental data. More transparent points corresponding to structures with lower statistical weights.

Histograms were estimated for each projection using 60 bins in each dimension. Data points belonging to histogram bins that contain at least one data point from each force field are colored dark gray. **B** Overlap of ELViM latent space kernel densities of unbiased and reweighted IDP ensembles. Values in the blue triangles reflect the ELViM density overlap of unbiased MD ensembles derived from different force fields, values in red triangles reflect the ELViM density overlap of reweighted ensembles derived from different force fields, and the diagonal elements reflect the ELViM density overlap of the reweighted ensemble derived from each force field with its parent unbiased MD ensemble.

the Supporting Information section "Assessing the accuracy of IDP conformational ensembles".

## Comparing IDP ensembles from different force fields

To assess the similarity of unbiased MD ensembles and the IDP conformational ensembles calculated in this work, we compare the populations of α-helical and β-sheet secondary structure elements, the populations of intramolecular contacts, and the free energy surfaces of each ensemble as a function of $R_g$ and the α-helical order parameter $S\alpha$[64] (see "α-helical order parameter $S\alpha$" in the "Methods" section of the Supporting Information). Comparisons of unbiased and reweighted ensembles are shown for Aβ40 (Fig. 1, Supplementary Figs. 4–6), ACTR (Fig. 2, Supplementary Figs. 8–10), drkN (Supplementary Figs. 12–15), PaaA2 (Fig. 3, Supplementary Figs. 17–19), and α-synuclein (Supplementary Figs. 21–24). A detailed discussion of the differences in the structural properties of unbiased and reweighted IDP ensembles is included in the Supporting Information section "Comparing IDP conformational ensembles derived from different force fields".

While comparing IDP ensembles using selected reaction coordinates of interest (such as $R_g$ and $S\alpha$) can be intuitive and informative, we seek a more objective and quantitative description of the similarity of unbiased and reweighted ensembles of IDPs. One possible approach for quantifying the structural similarity and population overlap of

dynamic biomolecules is to compare the distributions of dihedral angles alongside global geometric descriptors, as previously demonstrated for nucleic acids[65]. Due to the extreme flexibility and large number of uncoupled dihedral degrees of freedom in IDPs, IDP ensembles with similar dihedral distributions may not contain any conformations with similar atomic coordinates. As one application of our efforts to determine accurate atomic-resolution ensembles of IDPs is to obtain mechanistic insight into their binding mechanisms with small molecule drugs[6–11], we desire comparisons that are sensitive to the relative positions of backbone and sidechain atoms and the presence of long-range interactions. Accordingly, we choose to quantify the similarity of IDP conformational ensembles directly from atomic positions.

To enable quantitative comparisons of IDP ensembles from atomic positions, we utilize the energy landscape visualization method (ELViM) for dimensionality reduction[66,67]. The ELViM approach uses the distances between $C\alpha$ carbons in each conformation of a protein ensemble as an input, calculates a dissimilarity matrix between all pairs of conformations based on differences in the populations of $C\alpha$–$C\alpha$ contacts, and projects the information contained in the high-dimensional dissimilarity matrix onto a low-dimensional latent space. Projecting IDP ensembles onto a low-dimensional latent space derived directly from atomic coordinates provides a more objective

comparison of the similarity of ensembles than examining the similarity of a small number of subjectively selected structural descriptors, and the ELViM approach has been shown to be effective for comparing conformational ensembles of IDPs[66].

We apply the ELViM approach to compare the similarity of the conformational ensembles sampled in the unbiased a99SB-*disp*, C36m, and C22* MD simulations of A$\beta$40, ACTR, drkN SH3, PaaA2, and $\alpha$-synuclein. For each protein, we concatenated the three unbiased MD ensembles into a single merged ensemble, computed the dissimilarity matrix between all conformations, and used the ELViM algorithm to project the conformations of the merged ensemble onto a two-dimensional (2D) latent space (see "IDP ensemble comparisons" in the "Methods" section). We then used kernel density estimates of each unbiased and reweighted ensemble projected onto the ELViM latent space to compare the ensembles. A comparison of the projections of the unbiased and reweighted ensembles of A$\beta$40 and ACTR on their ELViM latent spaces is shown in Figs. 4, 5. We note that we utilize the same ELViM projections for unbiased and reweighted ensembles. Therefore, the locations of points in the ELViM projections do not change after reweighting; only their statistical weights change.

To obtain more insight into the nature of the ELViM latent spaces, we project the $R_g$ and $S\alpha$ values of each conformation of the merged ensembles on their respective ELViM latent space and present snapshots of conformations from selected regions of ELViM density projections in unbiased and reweighted ensembles. A comparison of the projections of the unbiased and reweighted ensembles of drkN SH3, PaaA2, and $\alpha$-synuclein on their respective ELViM latent space is shown in Supplementary Figs. 25, 26, and 27, respectively.

We display overlays of the ELViM latent space embeddings of all ensembles of A$\beta$40, drkN SH3, ACTR, PaaA2 and $\alpha$-synuclein in Fig. 6. To provide a quantitative measure of the similarity of the unbiased and reweighted ensembles of each protein in the ELViM latent space, we define a *density overlap* metric $S$ (Eq. 3), which is analogous to the overlap integral used to quantify the overlap of electronic wave functions in quantum mechanics (see "Methods" section). We first construct a kernel density of each ensemble in the ELViM latent space. We compare the similarity of two ensembles by computing the overlap integral $S$ of their kernel densities, normalized such that the value of $S$ ranges from [0,1] (Eq. 5). Two densities with no overlapping points will have an overlap integral value of $S = 0$, while two identical densities will have an overlap integral value of $S = 1$. We display the values of the overlap integrals between the reweighted and unbiased ensembles of each protein, expressed as an overlap percentage ($S \cdot 100\%$) in Fig. 6. Values in the blue triangles reflect the overlap of unbiased MD ensembles derived from different force fields, values in red triangles reflect the overlap of reweighted ensembles derived from different force fields, and the diagonal elements reflect the overlap of the reweighted ensemble derived from each force field with the unbiased MD ensemble from which it was derived. For comparison, we also performed the analysis described on projections using the conventional linear dimensionality reduction principal component analysis (PCA) (Supplementary Fig. 28). We find that we obtain consistent results using either dimensionality reduction technique.

The ELViM projections of A$\beta$40 and ACTR ensembles, shown in Figs. 4 and 5, illustrate that the unbiased a99SB-*disp*, C36m, and C22* MD simulations of these proteins sample the same regions of conformational space with different probabilities, and that these probabilities are adjusted to produce highly similar conformational distributions after reweighting with experimental data. The reweighted ensembles of ACTR and A$\beta$40 share similar high-density regions in the ELViM latent space, which is consistent with the similarity of the intramolecular contact maps and secondary structure propensities of the reweighted ensembles of these proteins (Fig. 1, Supplementary Figs. 6, 8–10). Unbiased drkN SH3 ensembles initially have relatively

little overlap in the ELViM latent space, but their overlap is substantially increased upon reweighting (Fig. 6, Supplementary Fig. 25).

The ELViM projections of PaaA2 ensembles shown in Fig. 6 and Supplementary Fig. 26 illustrate that there is very little overlap of unbiased MD ensembles derived from different force fields in the space of $C\alpha$ contacts. This is reflected by overlap percentages of less than 2% between unbiased ensembles (Fig. 6). As a result, while reweighting marginally increases the overlap of the ensembles, reweighted PaaA2 ensembles are substantially more similar to the unbiased MD ensembles from which they were derived than to one another. This results from the fact that the stable helical elements in PaaA2 ensembles can pack in distinct discrete orientations in the overly collapsed C22* and C36m ensembles, and the unbiased MD ensembles do not sample the same packing orientations. The unbiased and reweighted ensembles of $\alpha$-synuclein derived from different force fields are as similarly disjoint as the PaaA2 ensembles in the ELViM latent space, with almost no overlap before or after reweighting (Fig. 6, Supplementary Fig. 27). This results from the propensity of the C22* and C36m force fields to substantially underestimate the $R_g$ of of IDPs longer than 60 residues[19].

## Discussion

In this investigation, we have developed a simple, robust, and automated maximum entropy reweighting method to calculate accurate atomic-resolution conformational ensembles of IDPs using large datasets consisting of several different types of experimental data. Our proposed reweighting procedure contains a single free parameter, the desired effective ensemble size of the reweighted ensemble, and naturally balances the strengths of restraints for different types of experimental data. This approach does not require any a priori knowledge of the accuracy of an initial IDP ensemble, the magnitude of errors of experimental measurements, the correlation of experimental observables, or the accuracy of forward models for predicting experimental data. We demonstrate, through extensive cross-validation, that in favorable cases the proposed reweighting approach simultaneously improves agreement with several types of experimental NMR data and SAXS data with minimal overfitting while maintaining a desired degree of sampling of the most populated regions of conformational space in unbiased MD ensembles.

We caveat, however, that the success of the reweighting protocol proposed here, and all maximum-entropy reweighting protocols in general, is contingent on using reasonably accurate, well-sampled initial ensembles as input for reweighting[26]. If an initial unbiased ensemble does not sample an appreciable population of conformational states that are consistent with experimental data, no reweighting protocol will be able to correct this distribution to resemble a realistic solution ensemble.

The maximum entropy reweighting procedure developed in this investigation enables us to address the following question: Are modern force fields becoming sufficiently accurate that MD simulations with adequate sampling will converge to a similar underlying conformational distribution when extensive experimental datasets are used for reweighting? This question can be alternatively viewed as: with sufficient experimental data, does the problem of determining structural ensembles of IDPs with modern force fields and maximum entropy reweighting methods become well-defined?[12] In the case of A$\beta$40 and ACTR, two shorter IDPs (40 and 69 residues, respectively) with moderate populations of residual secondary structure elements, we find substantial overlap in the unbiased a99SB-*disp*, C36m and C22* MD ensembles from both traditional structural descriptors and from a low-dimensional projection of the ensembles onto a latent space defined by the positions of $C\alpha$ carbons. Upon reweighting, we find that the ensembles become highly similar to one another and have exceptional agreement with experimental data.

These encouraging results suggest that the accessible conformational space of these proteins is well sampled in 30 μs MD simulations run with different force fields, and that available experimental NMR and SAXS datasets provide sufficient information to correct the simulated ensembles into similar conformational distributions. While this is not evidence that these reweighted ensembles are perfectly faithful representations of the *true* solution ensembles of these proteins, this represents substantial progress in the field of IDP structure calculation; it is extremely encouraging that each force field does not produce a unique reweighted ensemble with equally good agreement with experimental restraints. These results suggest that with modern force fields and sufficient NMR and SAXS data, the challenge of determining structural ensembles of IDPs with fewer than 70 residues may now be becoming well-defined, in that consensus descriptions of solution ensembles are beginning to emerge.

In the case of drkN SH3, the unbiased ensembles appear relatively similar aside from substantial populations of β-sheets in the C36m and C22* ensembles. Reweighting reduces the populations of these β-sheets and substantially increases the overlap of the conformational ensembles, but agreement with experimental data indicates that the reweighted a99SB-*disp* drkN SH3 ensemble is the most accurate representation of this protein in solution. The reweighted ensembles of PaaA2 and α-synuclein produced in this work have extremely little overlap. The unbiased MD ensembles of these proteins appear to sample almost entirely disjoint regions of conformational space. The lack of overlap of unbiased ensembles is likely the result of a combination of force field inaccuracies and insufficient sampling in unbiased 30μs MD simulations. In particular, the C22* and C36m force fields have been found to substantially underestimate the $R_g$ of IDPs longer than 60 residues[19].

While reweighting improves some properties of the overly compact C22* and C36m ensembles of PaaA2 and α-synuclein, such as the distributions of backbone dihedral angles, we are clearly far from identifying a consensus description of the solution ensembles of these proteins. Based on the superior agreement with experimental data, the reweighted a99SB-*disp* ensembles of PaaA2 and α-synuclein are likely substantially more representative of the true solution ensembles of these proteins than the reweighted C22* and C36m ensembles.

Accurate, force field independent conformational ensembles of IDPs, like the ones determined here for Aβ40, ACTR, and drkN SH3, are valuable for several applications. These reweighted ensembles can be used as target distributions when assessing the accuracy of coarse-grained representations of IDPs[68,69] and machine learning methods for predicting atomic-resolution conformational ensembles[47,48] and ensemble properties[3,70] of IDPs. The ensembles calculated in this work will also be valuable for benchmarking future force field development efforts[17]. Additionally, reweighted ensembles calculated with the approach proposed here can serve as training data for machine learning approaches to generate IDP conformational ensembles at reduced computational cost compared to standard MD simulations. Force field independent ensembles could complement the information encoded in the co-evolution of amino-acid sequences and the structures deposited in the Protein Data Bank[71], on which methods like AlphaFold3[49] were trained, but which provide limited information about the conformational heterogeneity of dynamic and disordered proteins. A major challenge for these efforts will be generating a sufficient number of high-quality conformational ensembles to train IDP models that generalize well to unseen data, especially when considering the vast conformational space of IDPs.

In conclusion, we have proposed and validated a robust approach for integrating MD simulations of IDPs with extensive experimental sets to improve the accuracy of atomic-resolution conformational ensembles of IDPs. We have provided an in-depth comparison of the reweighted IDP ensembles derived from different force fields and have shown that in favorable cases these ensembles converge to very similar conformational distributions after reweighting. We have also shown that for several proteins, IDP ensembles obtained from reweighting atomic-resolution MD simulations have very similar properties to IDP ensembles calculated directly from experimental data with minimal ensemble methods. These results demonstrate substantial progress in the field of IDP ensemble modeling and suggest that the field may be maturing from the realm of assessing the accuracy of disparate computational models to the realm of atomic-resolution integrative structural biology. We anticipate that the maximum entropy reweighting protocol proposed here will provide a valuable tool for calculating accurate atomic-resolution IDP ensembles, assessing the accuracy of future force field development efforts, scrutinizing IDP ensembles obtained from artificial intelligence and machine learning methods, and ultimately providing high-quality data to train efficient and accurate deep learning models for IDP conformational ensemble generation.

## Methods

In our proposed maximum entropy reweighting approach, we extend the maximum entropy reweighting formalism of Bussi et al.[25–27]. We review this formalism in the Supporting Information "Theory" section, and describe modifications and extensions proposed in this work here.

### Determining regularization parameters

In the formalism of Bussi and coauthors[25–27], regularization parameters are introduced to account for systematic and random errors in experimental data and forward model models used to calculate experimental data from structural ensembles and to account for limited confidence in a simulated prior distribution. Errors are modeled by introducing an auxiliary variable for each experimental data point $\epsilon_i$, which represents expected differences between experimental and calculated values considering all possible sources of errors (SI Eq. 8) The expected distribution of values of $\epsilon_i$ for each experimental observable is modeled by a Gaussian function with a standard deviation defined by the regularization parameter $\sigma_i$ (SI Eq. 9). The value of $\sigma_i$ corresponds to the level of confidence in the $i^{th}$ experimental data point and ultimately determines how well the reweighted ensemble will fit the data. $\sigma_i = \infty$ implies no confidence in the data and results in a reweighted ensemble that is identical to the unbiased simulation. $\sigma_i = 0$ implies complete confidence in the data and results in a reweighted ensemble in which the $i^{th}$ data point is exactly matched. The selection of appropriate values of the regularization parameters $\sigma_i$ for each experimental data point is therefore essential to balance one's confidence in a prior model (i.e., the accuracy of a force field), error estimates of experimental measurements, and error estimates of the accuracy of forward models for predicting experimental data. Ultimately, the properties of the resulting conformational ensembles will be highly dependent on the relative magnitudes of the regularization parameters selected for each experimental data point.

Here, we assign one regularization parameter $\sigma_i$ to each data point. To optimize the grid search over the space of $\sigma_i$, we decompose this parameter into two contributions:

$$\sigma_i = \sqrt{\sigma_{i,MD}^2 + \sigma_{reg}^2}. \tag{1}$$

where $\sigma_{reg}$ is the regularization parameter, one per data type, that describes both experimental and forward model errors of this specific type of experiment. The $\sigma_{i,MD}$ parameter, one per data point, represents both the statistical errors of forward models to predict experimental data and statistical errors that result from calculating average quantities over a finite-size ensemble. We calculate $\sigma_{i,MD}$ using the Flyvbjerg block analysis[63,72] technique to account for the correlation between adjacent frames in the MD simulation. Alternatively, $\sigma_{i,MD}$ can be estimated as the standard error of the mean for uncorrelated data. We use a single free parameter, the Kish

ratio (Eq. 2), to determine the optimal values of $\sigma_{reg}$ for each experimental data type (*vide infra*).

From a conceptual perspective, we view $\sigma_{MD}$ and $\sigma_{reg}$ as regularization terms that are sufficiently large to account for additional error terms that are not individually represented with their own separate regularization parameters (including random and systematic experimental errors). We emphasize that for the classes of experimental NMR and SAXS data considered in this work, reported magnitudes of experimental errors in experimental databases are substantially smaller (in many cases over an order-of-magnitude smaller) than the magnitudes of our regularization terms $\sigma_{MD}$ and $\sigma_{reg}$ used in this investigation and in these cases explicit inclusion of experimental error estimates with additional regularization parameters has a negligible effect on reweighted ensembles.

### Kish ratio

Maximum entropy reweighting algorithms change the statistical weights of each conformation of the input ensemble in a minimal way to improve the agreement with the available experimental data. In a conformational ensemble derived from an unbiased MD simulation with $N$ frames, each conformation will have an equal statistical weight of $1/N$: $\boldsymbol{w_{MD}} = \{1/N, 1/N, ..., 1/N\}$. Upon reweighting, the weights of all frames are modified to $\boldsymbol{w} = \{w_0, w_1, ..., w_N\}$. If the initial unbiased MD ensemble is already in excellent agreement with all the experimental data used for reweighting, it may be possible to satisfy these data with very small perturbations to the weights of each frame. In this instance, the weight of most frames will still be close to $1/N$. If instead a large fraction of frames of the unbiased MD ensemble are in poor agreement with the experimental data, in order to satisfy the data within a desired threshold, it may be necessary to reduce the statistical weights of a significant number of frames to values close to 0, effectively discarding these frames from the reweighted ensemble.

To quantify the overall change in weights that occurs upon reweighting, we use the (normalized) Kish effective sample size[27,50], or *Kish ratio K*:

$$K = \frac{1}{N} \left( \sum_{i=1}^{N} w_i \right)^2 / \sum_{i=1}^{N} w_i^2 \qquad (2)$$

where $w_i$ is the value of the statistical weight of the $i$th frame. The Kish ratio is a measure of the dimension of the reweighted ensemble expressed in terms of the fraction of frames of the original ensemble with significant statistical weight after reweighting. A reweighted ensemble where ~50% of the frames have statistical weights close to zero will have a Kish ratio $K \approx 0.50$. Conversely, if an unbiased MD simulation has 10,000 frames, a reweighted ensemble with a Kish ratio $K = 0.01$ would have approximately 100 frames (1% of frames) with statistical weights substantially larger than 0. By definition, a conformational ensemble derived from an unbiased MD simulation where all $N$ weights are equal has a Kish ratio is $K = 1.0$.

As discussed in the previous section, the $\sigma_i$ parameters express our confidence in the experimental data, and the weights assigned to each simulated conformation are highly dependent on the chosen values of $\sigma_i$. Depending on the quality of the original MD ensemble, reweighting with a given value of $\sigma_i$ can lead to dramatically reducing the Kish ratio of the reweighted ensemble, thus compromising the statistical accuracy of any average property calculated over the reweighted ensemble. Reweighted ensembles with small Kish ratios may effectively discard all structures representative of free energy basins in an initial MD simulation, and satisfy experimental restraints with a small set of rarely populated conformations (Supplementary Fig. 2). It is therefore desirable to ensure that the reweighted ensemble

preserves a sufficiently large Kish ratio in order to be able to make predictions of the properties of IDPs, such as the relative positions and orientations of sidechain residues, with a desired statistical accuracy.

### Maximum entropy reweighting with a single free parameter

Our approach to combine multiple types of experimental data to reweight IDP ensembles uses only one free parameter: the Kish ratio of the reweighted ensemble (Eq. 2). Our protocol proceeds as follows:

1. We calculate $\sigma_{i,MD}$ for each data point from the unbiased MD simulation using a Fylvberg[63] blocking analysis.
2. For each experimental data type, we perform reweighting at a wide range of values of $\sigma_{reg}$ and monitor the Kish ratio of the reweighted ensembles. For each data type, we choose a value of $\sigma_{reg}$ based on a selected Kish ratio threshold. Here, we select the minimum value of $\sigma_{reg}$ that produces an ensemble with Kish ratio $K \geq 0.10$ for each data type. This procedure establishes the relative values of $\sigma_{reg}$ for each data type (i.e., $\sigma_{reg-C\alpha}$, $\sigma_{reg-J_{HN H\alpha}}$, $\sigma_{reg-SAXS}$, etc.).
3. A global reweighting is performed using all experimental data as restraints. In this step, we determine a global scaling factor $\sigma_{reg-Global}$ for the values of $\sigma_{reg}$ calculated for each data type in step 2. We perform reweighting at a range of values of $\sigma_{reg-Global}$ and monitor the Kish ratio of the reweighted ensembles. We select the minimum value of $\sigma_{reg-Global}$ that produces a final reweighted ensemble with Kish ratio $K \geq 0.10$.

### Normalized RMSE (RMSE$_N$) comparisons

To compare the relative accuracy of ensembles before and after reweighting, we introduce an expression for the normalized RMSE (RMSE$_N$), where we normalize the value of the RMSE calculated for each experimental data type by the corresponding value in the unbiased a99SB-*disp* MD ensemble (RMSE$_N$ = RMSE/RMSE$_{a99SB-disp}$).

To compare reweighted ensembles obtained from different force fields, we define a global quality index for each system: the averaged normalized RMSE ($\overline{RMSE_N}$). For each force field and data type $i$, we compute the agreement between calculated and experimental data in the reweighted ensemble (RMSE$_i$). We then normalize each value of RMSE$_i$ by the corresponding RMSE value in the unbiased a99SB-*disp* MD ensemble (RMSE$_{N,i}$ = RMSE$_i$/RMSE$_{i,a99SB-disp}$) and finally average across all M data types to obtain $\overline{RMSE_N} = 1/M \sum_{i=1}^{M} RMSE_{N,i}$.

### IDP ensemble comparisons

We provide a quantitative measure of the structural similarity of conformational ensembles by collectively analyzing a set of molecular dynamics simulations of the same protein by using the energy landscape visualization method (ELViM)[66,67] for dimensionality reduction and computing the overlap of latent space densities.

For each protein, we first down-sampled each trajectory by a factor of two, so that each ensemble contained 14988 frames. We then concatenated all three unbiased trajectories into a single merged ensemble, which contained a total of 44964 frames. We used ELViM to compute a dissimilarity matrix for this merged ensemble with the hyperparameters $\sigma_0$ and $\epsilon$ set to 1 and 0.15, respectively, and projected the conformations of the merged ensemble onto a 2D latent space. Individual kernel densities were calculated for each simulation dataset (each unbiased and reweighted ensemble of each protein) and sampled over the global extrema in each dimension of the ELViM latent space using an 80 by 80 grid. Gaussian kernel densities were estimated using Scott's rule[73] to determine kernel bandwidths for both reweighted and unbiased kernel density estimates. For a pair of kernel densities, $D_1$ and $D_2$, we define the overlap integral, $S$:

$$S = N_1 N_2 \int \int D_1(x_1, x_2) D_2(x_1, x_2) dx_1 dx_2 \qquad (3)$$

where $N_1$ and $N_2$ are normalization constants for the distributions $D_1$ and $D_2$, respectively, chosen such that the integrand is defined in the range: [0, 1]. For a kernel density $D_i$, the normalization constant is defined as :

$$N_i = \frac{1}{\sqrt{\int \int D_i(x_1, x_2) D_i(x_1, x_2) dx_1 dx_2}} \tag{4}$$

If evaluations of kernel density estimators are compiled into vectors $\boldsymbol{D_1}$ and $\boldsymbol{D_2}$, this computation is equivalent to the normalized dot product.

$$S = \frac{\langle \boldsymbol{D_1} | \boldsymbol{D_2} \rangle}{\| \boldsymbol{D_1} \| \| \boldsymbol{D_2} \|} \tag{5}$$

Using the definitions in Eq. 3–5, $S$ is a positive value from [0,1]. Two kernel densities with no overlapping points have an overlap of value of $S = 0$, while two identical kernel densities have an overlap of $S = 1$. The values of overlap integrals $S$ can be converted to a percentage by multiplying $S \cdot 100\%$. We computed the overlap integral of the ELViM projections of unbiased and reweighted ensembles derived from simulations of each protein with each force field, and compare the overlap of the ELViM projections for all pairs of unbiased and reweighted ensembles for each protein (Fig. 6). We repeat this analysis on projections obtained from linear dimensionality reduction technique PCA in Supplementary Fig. 28. We find that we obtain consistent results using either dimensionality reduction technique.

We note values of the overlap integral $S$ are sensitive to the width of the Gaussian kernel densities used for kernel density estimation (KDE). For comparison, we also computed overlap integrals using histograms over a range of bin sizes. We find that $S$ values computed with KDEs with bandwidths estimated by Scott's rule[73] are similar to $S$ values calculated using histograms when the number of histogram bins does not dramatically exceed the number of conformations in reweighted ensembles.

We note that when computing the overlap integral $S$ between a reweighted ensemble and the original unbiased ensemble, the square of the overlap integral $S$ converges to the Kish score in the limit of infinitely many bins. It is, however, uninformative to compare ensembles derived from independent simulations when the number of histogram bins dramatically exceeds the number of conformations being compared, as it is highly unlikely that two *identical* conformations will be sampled in independent simulations. As the number of histogram bins becomes large, all conformations sampled in independent simulations will be assigned to unique bins, and the overlap $S$ approaches zero regardless of the similarity of the conformations sampled.

### Reporting summary
Further information on research design is available in the Nature Portfolio Reporting Summary linked to this article.

## Data availability
All experimental data used to reweight MD trajectories, the predicted values of experimental data calculated from each trajectory, and values of the structural descriptors used to compare ensembles are freely available from the GitHub repository (https://github.com/paulrobustelli/Borthakur_MaxEnt_IDPs_2024/) and Zenodo (https://doi.org/10.5281/zenodo.16814130). Reweighted ensembles of each protein have been deposited to the protein ensemble database under the following deposition codes: Aβ40: [PED00531], [PED00532], [PED00533]. ACTR: [PED00534], [PED00535], [PED00536], drkN SH3: [PED00537], [PED00538], [PED00539], PaaA2: [PED00540], [PED00541], [PED00542] α-synuclein: [PED00543], [PED00544], [PED00545]. The NMR chemical shifts used for reweighting of ACTR

and PaaA2 trajectories were obtained from previously reported experiments and are deposited in the Biological Magnetic Resonance Bank entries [BMR15397] and [BMR18841], respectively. Reweighted ensembles are also provided in the accompanying GitHub and Zenodo repositories. Source data are provided with this paper as Source Data File. All previously reported[19] MD trajectories analyzed in this work are available for non-commercial use by request from D.E. Shaw Research (Trajectories@DEShawResearch.com). Source data are provided with this paper.

## Code availability
All code used to calculate experimental data from MD trajectories, perform reweighting, analyze ensembles, and compare energy landscape visualization method (ELViM) projections[66,67] is freely available from the Github repository [https://github.com/paulrobustelli/Borthakur_MaxEnt_IDPs_2024/] and Zenodo [https://doi.org/10.5281/zenodo.16814130]. The ELViM code was adapted from the GitHub repository [https://github.com/VLeiteGroup/ELViM/].

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

## Acknowledgements
P.R., K.B., and T.R.S. acknowledge the support of NIH award R35GM142750. M.B. acknowledges funding from the European Research Council (ERC) under the European Union's Horizon 2020 research and innovation programme ("bAIes" ERC grant agreement no. 101086685). M.B. and F.P.P. acknowledge the support of the French Agence Nationale de la Recherche (ANR), under grant ANR-20-CE45-0002. The authors thank Vincent Schnapka, Vitor Leite, Giovanni Bussi, and Ivan Gilardoni for useful discussions and comments.

## Author contributions
P.R. and M.B. conceived, designed, and supervised the research. K.B., T.R.S., and F.P.P. performed research. F.P.P. performed initial reweighting calculations. K.B., M.B., and P.R. designed the final reweighting protocol. K.B. performed all reweighting and ensemble analysis calculations. T.R.S. designed and performed calculations for dimensionality reduction and latent space comparisons. K.B. and P.R. drafted the manuscript. K.B., T.R.S., M.B., and P.R. revised and edited the manuscript.

## Competing interests
The authors declare no competing interests.
