## [Transparent Peer Review file · Nature Communications]

Determining accurate conformational ensembles of intrinsically disordered proteins at atomic resolution

Corresponding Author: Professor Paul Robustelli

Version 0:

Reviewer comments:

Reviewer #1

(Remarks to the Author)

In their manuscript "Determining accurate conformational ensembles of intrinsically disordered proteins at atomic resolution" by Borthakur et al., the authors tackle the important task of providing atomistic ensembles of intrinsically disordered proteins (IDPs). They do so by reweighting ensembles to match experimental data.

The paper is well written and the graphical representations of the results are well done.

Their main result is the comparison of three widely-used atomistic force fields. They use the simulation data of a previous study, where the three force fields have been applied to five widely-studied IDPs. Both, the previously published molecular dynamics trajectories and the experimental data are extensive. The authors compare ensembles for different force fields with each other and before and after reweighting with the experimental data. The authors find that the force fields work well for three IDPs, leading to similarly reweighted ensembles. For two of the five IDPs, this is not the case, with one force field performing clearly better than the others.

These results are highly interesting to force field developers and for practitioners using molecular dynamics simulations and who have to choose an appropriate force field. I would not be surprised if among available IDP ensembles, the provided ensembles are the closest to the "ground truths".

To properly assess the meaning and implications of their results, I would like the authors to respond to the following points. Below these, please find a list of additional comments.

(1) Overfitting?

One main challenge in ensemble refinement is to avoid overfitting of the experimental data. Overfitting can only be properly assessed, if experimental errors are included. As I understand, the authors do not use experimental errors in their approach.

For example, it might be that the unbiased simulations already fit some of the experimental data perfectly within these errors. Their approach of scanning their regularization parameter σ_{reg} with decreasing Kish sampling size might give lower RMSE also in such cases, which would be clearly overfitting. How is this avoided? Or even before that, how does one quantify overfitting in their approach?

(2) Role of regularization parameters?

One major point by the authors is that they have developed a procedure to estimate regularization parameters to weigh the different experimental data types, independent of the actual experimental errors. They do estimate the sampling errors in molecular simulations, σ_{MD} , though. I am wondering if estimating σ_{MD} is even necessary. Would the authors get the same results using their procedure if all σ_{MD} were set to zero?

If σ_{MD} are zero, then their regularization parameter σ_{reg} as well as their global rescaling factor $\sigma_{\text{reg-global}}$ would enter a multiplicative parameter for each data type. If so, this distinguishing property of their approach from previous approaches would not be needed. If this is not the case and their approach of using σ_{MD} explicitly has clear

advantages, then this should be clarified and best illustrated with reweighting results.

Specifically, in the L-curve analysis mentioned by the authors, one uses a single multiplicative factor for the entropy. Decreasing this parameter decreases the Kish sampling size. In principle, one could pick the value of this factor that gives the desired Kish sampling size. Importantly, this parameter has the effect of a global multiplicative parameter for the total errors of the experimental data, similar to $\sigma_{\text{reg-global}}$ if $\sigma_{\text{MD}}=0$. One would assume that using multiplicative scaling factors for individual data types, determined in a procedure similar to what the authors do, and using one global scaling factor will have a similar effect than their approach.

(3) Best possible ensembles?

The authors chose to set a value for the Kish sampling size to compare force fields, which is fair. When it comes to the quality of the ensembles, fixing the sampling size for all force fields is expected to lead to sub-optimal ensembles for individual force fields. The Kish sampling size for the best force field might be smaller than necessary and (some of!) the experimental data might be overfitted, when taking into account experimental errors. For the worst force fields, agreement with the experimental data could perhaps be improved by lowering the Kish sampling size.

Although the authors provide good quality ensembles, I am wondering why not go the extra step and provide the best possible ensembles for each protein? After all, using one value of the Kish sampling size for all simulations to compare the ensembles puts a constraint on the quality of the reweighted ensemble.

For example, one could refine the combined ensembles for all force fields of a single protein into one and refine it against the experimental data. Their regularization approach should work. Force fields could have different initial probabilities, each of which has uniform initial weight.

(4) Central research question?

I do think that the central research question by the authors “with sufficient experimental data, can we determine physically realistic atomic-resolution IDP ensembles with conformational properties that are independent of the force fields used to generate the initial computational models?” can be easily tackled. I assume the authors agree.

For example, consistency between force fields could just mean that they do the same, not necessarily that they do it correctly. Similarly, having agreement with experiment does not tell us how informative experimental data are when it comes to structural ensembles. Also, the issue of overfitting remains. In my opinion, the authors gave answers to the questions (1) How good are modern force fields when it comes to simulating IDPs? and (2) How can we reweight consistently when we want to compare force fields without using experimental errors? Although a step in the right direction, their results fall short of answering their central research question, in my opinion.

Additional comments in arbitrary order

- I think readers could benefit from more guidance when it comes to the energy landscape visualization method (ELViM), used prominently by the authors. The energy landscape visualization method is not (yet?) an established method. To understand the plots, one has to have some minimal understanding of this method: In this method, each point in the 2D projection corresponds to a structure in the ensemble. Points are arranged such that the distance between points correlates with the dissimilarity of these two structures. Similar structures are close-by and form density peaks. Consequently, individual projections are not comparable. To overcome this problem, the authors combine the ensembles they want to compare in a single ensemble for which they then perform ELViM. They then separate the individual ensembles in these projections for analysis, e.g., for calculating the overlap integral. I do not know if their approach is original, already contained in the ELViM paper, or in the paper they have been inspired by (Reference 11 of the manuscript). Anyhow, I think this important point deserves clarification.
- Are error bars available for SAXS data (e.g., for ACTR in Figure 2 A)? SAXS experiments on IDPs are difficult and SAXS intensities are often not that informative. Usually, the radius of gyration is the most important experimental information but often contaminated by aggregation. At higher angles, SAXS intensities of IDPs are usually quite featureless. Without error bars it is impossible to judge even visually the quality of the data and thus the quality of the fits. This question relates again to overfitting.
- In Fig. 3 A, the SAXS data seems to have a significantly different radius of gyration than the reweighted ensemble for a99SB-disp. The fit is even worse in SI Fig. 12. Are calculated radii of gyration significantly different from the experimental data? Without error bars, it is hard to judge how to think about these fits.
- I am wondering about the systematic errors in the calculations of the experimental observables (see, for example, Frohliking et al. JCP 158 2023). These calculations depend often on the choice of parameter values, which themselves are uncertain. This uncertainty implies systematic errors. I assume these parameter uncertainties can have dramatic effects on the quality of the reweighting.
- For example, when the authors fit SAXS intensities, do they use a scaling parameter and a shift parameter for the intensities in the fitting? The shift parameter not only corrects for errors in the background subtraction but also accounts for contrast differences.

- The authors mention that they follow the formalism of Bussi et al. If I am not mistaken, Bussi et al. have shown in the supporting information of Cesari et al. JCTC 2016 that for Gaussian noise, which the authors use here, their approach is equal to the approach of Hummer and Kofinger JCP 2015. If so, shouldn't this be clearly pointed out? In the field of ensemble refinement, many approaches that superficially seem different actually are the same at the core. Being clear about it benefits the field and possible new users, in my opinion.
- Do the authors have some data or rationale to corroborate that ensembles are well-sampled in simulations? Sampling must depend on chain length and might depend on secondary structure content, for example. The quality of sampling must be different for different IDPs. It would be great to get to know more about the quality of sampling, when it comes to different structural properties. Block averaging does not give all the information needed to judge how well some properties have been sampled.
- I am not sure that the emphasis on "a single free parameter" gives the right impression. Other methods also have a single free parameter, but rely on estimates of the theoretical and experimental errors. If I understand the approach of the authors correctly, it is more about being able to refine without knowing experimental errors.

(Remarks on code availability)

Reviewer #2

(Remarks to the Author)

In this paper, the authors integrated all-atom MD simulations with experimental data, namely NMR and SAXS using automated maximum entropy reweighting procedure. They applied this method to five well-studied intrinsically disordered proteins and obtained consistent structure ensembles from MD simulations based on different force-field parameters.

The methods are successful for three IDPs, while the results of the other two IDPs are not so impressive because of the separation of the unbiased MD ensembles with different force fields. These two are larger proteins, and enhanced conformational sampling in unbiased MD simulations is critical in the method. Although they used 30-microsecond MD simulations, which seem to be sufficiently long in the current standard of the conventional MD simulations, they are not enough for reliable reweighted structural ensembles.

Therefore, I think some of the sentences in this paper seem to be overstated.

In Abstract, "We demonstrate that when this approach is applied with sufficient experimental data, IDP ensembles derived from different MD force fields converge to highly similar conformational distributions."

In Results on page 8, "The conformational ensembles calculated in this work are in exceptionally good agreement with extensive experimental NMR and SAXS datasets, that describe both local and global structural properties of IDPs, and are in substantially better agreement with experimental data than the most accurate unbiased MD ensembles of these proteins reported in previous benchmark studies."

In Discussion on page 12, "We demonstrate, through extensive cross-validation, that the proposed reweighting approach simultaneously improves agreement with several types of experimental NMR data and SAXS data with minimal overfitting while maintaining a desired degree of sampling of the most populated regions of conformational space in unbiased MD ensembles."

These statements are valid only when sufficient conformational space is explored in unbiased MD simulations. I strongly suggest the authors change these statements and add the limitations of these methods.

The limitation highlights the importance of enhanced conformational sampling methods and/or multi-scale MD simulations to explore wide conformational spaces of intrinsically disordered proteins. Although the authors discussed enhanced sampling in the main text, they did not cite critical studies on developing enhanced conformational sampling methods. I suggest the authors add more citations to those works.

The energy landscape visualization method (ELViM) is used for dimensionality reduction. It looks interesting; however, the authors did not compare the performance and advantages of ELViM with other dimensionality reduction methods, such as PCA, tICA, UMAP, etc. The authors need to add more discussion about the comparison and advantages of ELViM.

In Figure 3A, I cannot find lines for C22*/TIP3P. Please check this figure carefully.

(Remarks on code availability)

They provided the code used in this study appropriately.

Reviewer #3

(Remarks to the Author)

In the manuscript entitled "Determining accurate conformational ensembles of intrinsically disordered proteins at atomic resolution", Kaushik Borthakur, et al., present a maximum entropy reweighting scheme to obtain conformational ensembles of intrinsically disordered proteins (IDPs) which match experimental observables such as SAXS or NMR data. The unbiased ensembles are obtained by long (30 μ s) atomistic molecular dynamics simulations with different force fields.

The method presented and its application to IDPs is timely and relevant. Although a conceptually similar method has been presented by Köfinger and Hummer (J. Chem. Phys. 160, 114111 (2024), doi: 10.1063/5.0189901) as also mentioned by the authors, the presented test cases are more complex and take more experimental data into account.

The manuscript is potentially suitable for publication in Nature Communications after major revisions. My points are given below:

1) Out of the five test cases discussed, three result in a substantial improvement of the conformational ensembles, while the conformational ensembles of the different force fields remain distinct for two test cases. This dependence on the performance of an atomistic force field is a strength (considering e.g. side chain orientations), but also a serious limitation of the method. This limitation should be discussed clearly in the manuscript.

2) Two out of the five test cases (PaaA2 and α -synuclein) do not show good agreement of the conformational ensembles between the different force fields and thus seem not to converge. Can the authors provide more details on the biophysical properties of these two test cases compared to the ones where the method was applied successfully? Why do the atomistic force fields used fail to produce sufficient number of similar conformations, which could be emphasized during the reweighting? Can the authors give some guidelines on the biophysical properties of the IDPs which might be favorable for the employed force fields?

3) The authors hypothesize that longer sampling of the MD simulations could improve the conformational ensembles. To get a better agreement for the two problematic test cases, could it be also helpful to have access to more and/or different experimental data?

4) How does the method perform for (larger) intrinsically disordered regions on structured proteins? Adding more test cases in this direction would be helpful for the community and emphasize the broad applicability of the presented maximum entropy reweighting protocol.

5) Figure 6A compares the energy landscape visualization method (ELViM) projections of the unbiased and reweighted MD ensembles. In case of the reweighted ensembles (bottom row), bins which at least one data point from each force field are colored dark grey. How does this analysis look for the unbiased ensembles? Please include this analysis and discuss the improvement due to reweighting.

6) The method itself is only introduced in the Supplementary Information (SI). However, it is also part of the results in my view, and an overview should be given to the reader in the main manuscript. In the current version of the manuscript (particularly in the first part of the Results section), many references to sections in the SI are given, so that the text does not read very well.

7) The α -helical order parameter S_{α} is not labelled consistently, with or without subscript.

8) More details for reference 50 should be provided.

(Remarks on code availability)

Version 1:

Reviewer comments:

Reviewer #1

(Remarks to the Author)

I thank the authors for their response and modification to the manuscript. Overall, I am content with their answers and changes. However, I find some of the responses partially superficial and I do not completely agree with everything that was said by them. I refrain from going into details.

Instead, reading their rebuttal and the concerns of the other reviewers, I would like to share my thoughts on the significance of their results. Unfortunately, I find neither the method nor the results convincingly significant.

From the details and the results, it is not clear to me if the progress of the method is truly substantial. For example, I think I understand the attractiveness of the Kish sample size. The authors fix the Kish sampling size in their procedure. They could also have fixed the related Kulback-Leibler divergence instead, as has been done previously. Both quantities quantify the difference between initial and reweighted ensemble. I cannot judge if the strengths of the Kish sampling sizes outweighs

their disadvantages compared to the Kullback-Leibler divergence. Although only a detail, it emphasizes the recipe character of their method, in my opinion. Also, I fail to understand why using a "single free parameter" is a hallmark of their method given the methods in the literature. That their methods is "simple" is in the eye of the beholder. A method being "robust" is important, but this is also not a unique property of their approach.

I am also not confident that the quality of the reweighted ensembles is a significant enough improvement. One could say that the authors prefer consistency between ensembles over quality of the ensembles, which is fine. However, their goal of having consistent ensembles for different force fields does not align with the goal of having the best possible ensembles. For example, with the limited information provided by the authors, we have to assume that the initial ensembles suffer from limited sampling. As pointed out by Reviewer 3, sampling could be improved by biasing, e.g., using the experimental data themselves. Such enhanced ensembles should improve the quality of the reweighted ensemble substantially. As the authors have pointed out, such biased sampling is challenging. Without it, though, the reweighting is not as useful as it could be. The provided ensembles are still useful though, perhaps the best atomistic ensembles of IDPs available. However, I personally would only use them with a healthy dose of skepticism.

In summary, I do not see sufficient significance of either their method nor their reweighted ensembles. I simply could not be convinced by the manuscript and the rebuttal. From the method point of view, it appears to me that their method is more like a recipe than a fundamental development. As we know, it is difficult to convince others of the advantages of their personal recipe. From the biological results point of view, not knowing how good their reweighted ensembles actually are strongly limits their usefulness. As we are all aware, assessing the quality of ensembles is intrinsically hard. We only know ensemble could be better.

I am aware that my response is not as positive as the authors must have hoped for. Please keep in mind that I do think that their work is important and interesting. I am just not convinced it is significant enough progress. Luckily, there are two other reviewers and an editor who might think differently. Good luck!

(Remarks on code availability)

Reviewer #2

(Remarks to the Author)

All reviewers pointed out the advantages and limitations of this method, particularly issues arising from the narrowness of the simulation search space. The authors carefully addressed these points in the revised manuscript, which left a favorable impression. In this sense, the revised manuscript can be significantly improved.

(Remarks on code availability)

Reviewer #3

(Remarks to the Author)

I thank the authors for satisfyingly addressing all my points raised and now support publication.

(Remarks on code availability)

Version 2:

Reviewer comments:

Reviewer #3

(Remarks to the Author)

In my view, the authors addressed the concerns of reviewer 1 mostly sufficiently.

There are aspects which I would like to comment on. The authors write in their rebuttal that main goal #1 is to provide a tool for the structural bioinformatics and computational biophysics communities to calculate meaningful ensembles of IDPs. However, this tool is based on long MD simulations which are not easy to obtain. This limitation prevented the authors from including further test cases of systems combining folded and disordered domains, as suggested by this reviewer. Thus, I anticipate that it will be challenging for the communities to employ the tool on a regular basis.

In contrast, main goal #2, convergence of IDP ensembles from different force fields using experimental data was achieved for 3 out of 5 cases. This is also acknowledged by reviewer 1.

While the method/tool might be similar to other strategies to refine ensembles using experimental data, the use seems to be possible also for non-experts if there are long enough MD simulations available. It has been applied by the authors to converge 3 IDP ensembles generated with different force fields, which is an application going beyond the test cases used for

comparable methods.

(Remarks on code availability)

We thank the reviewers for their detailed constructive feedback. We have updated the manuscript to address their concerns, including, new analyses and additional discussion and clarification in the text.

Included below is a point-by-point response to each reviewer's comments.

We hope the revised manuscript is now suitable for publication in Nature Communications.

REVIEWER COMMENTS

Reviewer #1 (Remarks to the Author):

In their manuscript "Determining accurate conformational ensembles of intrinsically disordered proteins at atomic resolution" by Borthakur et al., the authors tackle the important task of providing atomistic ensembles of intrinsically disordered proteins (IDPs). They do so by reweighting ensembles to match experimental data.

The paper is well written and the graphical representations of the results are well done.

Their main result is the comparison of three widely-used atomistic force fields. They use the simulation data of a previous study, where the three force fields have been applied to five widely-studied IDPs. Both, the previously published molecular dynamics trajectories and the experimental data are extensive. The authors compare ensembles for different force fields with each other and before and after reweighting with the experimental data. The authors find that the force fields work well for three IDPs, leading to similarly reweighted ensembles. For two of the five IDPs, this is not the case, with one force field performing clearly better than the others.

These results are highly interesting to force field developers and for practitioners using molecular dynamics simulations and who have to choose an appropriate force field. I would not be surprised if among available IDP ensembles, the provided ensembles are the closest to the "ground truths".

We thank the reviewer for these positive comments.

To properly assess the meaning and implications of their results, I would like the authors to respond to the following points. Below these, please find a list of additional comments.

(1) Overfitting?

One main challenge in ensemble refinement is to avoid overfitting of the experimental data. Overfitting can only be properly assessed, if experimental errors are included. As I understand, the authors do not use experimental errors in their approach.

For example, it might be that the unbiased simulations already fit some of the experimental data perfectly within these errors. Their approach of scanning their regularization parameter σ_{reg} with decreasing Kish sampling size might give lower RMSE also in such cases, which would be clearly overfitting. How is this avoided? Or even before that, how does one quantify overfitting in their approach?

We thank the reviewer for raising the concerns of the possibility of overfitting when not explicitly including a term for experimental errors.

In our initial manuscript, we had neglected to emphasize that in virtually all instances, the magnitudes of experimental errors are substantially smaller than the magnitudes of our regularization terms S_{MD} and S_{reg} and often over an order magnitude smaller than the agreement between calculated and experimental values in the final reweighted ensembles.

In **Review Response Table 1** (below), we compare the experimental error estimates of NMR chemical shifts obtained from the BMRB deposition (BMRB 18841) with the values of σ_{MD} , σ_{reg} and the final RMSE after reweighting obtained from reweighting the a99SB-disp PaaA2 trajectory.

Atom Type	Exp Error (ppm)	σ_{MD} (Average over all data points)	σ_{reg} (one per Atom Type)	$(\sigma_{\text{MD}}^2 + \sigma_{\text{reg}}^2)^{0.5}$ (Average over all data points)	Reweight RMSE (ppm)
H α	0.008	0.028	0.40	0.40	0.10
H	0.008	0.051	0.60	0.60	0.15
C α	0.04	0.193	2.01	2.02	0.58
C β	0.04	0.106	1.01	1.02	0.39
C	0.04	0.15	1.71	1.72	0.56
N	0.04	0.29	2.01	2.03	0.84

Review Response Table 1: Comparison of magnitudes of estimated experimental errors in NMR chemical shifts, values of regularization parameters σ_{MD} and σ_{reg} and the RMSE of the final reweighted ensembles for the reweighted a99SB-disp PaaA2 ensemble.

We note that the size of the experimental error estimates obtained from the experimental NMR chemical shift BMRB deposition is approximately an order of magnitude smaller than the RMSE between calculated and experimental shifts, indicating that we are not over-fitting data to within experimental error estimates. We find this to be the case in virtually all cases where explicit experimental errors are reported.

We note that for the SAXS and NMR data used for reweighting in this manuscript, the errors of forward model predictions are generally an order of magnitude larger than reported experimental errors. In additional controls (data not shown) we find that if we include an additional explicit error term σ_{exp} for each datapoint and calculating the final regularization parameter as $\sigma_i = (\sigma_{\text{exp},i} + \sigma_{\text{MD},i} + \sigma_{\text{reg}})^{0.5}$ we see no appreciable difference in the properties of the reweighted ensemble.

Based on these comparisons, we conclude that our proposed reweighting procedure is not overfitting to within experimental errors. From a conceptual perspective, we view σ_{MD} and σ_{reg} as terms that are sufficiently large to also account for error terms that are not individually represented with their own separate regularization parameters (including random and systematic experimental errors as well as random and systematic errors in forward model predictors). **We have updated the text to clarify this point:**

Updated text (Main Text Page 16):

From a conceptual perspective, we view σ_{MD} and σ_{reg} as regularization terms that are sufficiently large to account for additional error terms that are not individually represented with their own separate regularization parameters (including random and systematic experimental errors). We emphasize that for the classes of experimental NMR and SAXS data considered in this work, reported magnitudes of experimental errors in experimental databases are substantially smaller (in many cases over an order-of-magnitude smaller) than the magnitudes of our regularization terms σ_{MD} and σ_{reg} used in this investigation and explicit inclusion of experimental error estimates with additional regularization parameters has a negligible effect on reweighted ensembles.

Ultimately, however, we believe the strongest evidence for a lack of overfitting are the extensive leave-one-out cross-validation tests we performed in the manuscript, discussed in the SI section “A maximum entropy reweighting protocol with a single free parameter improves agreement with extensive experimental datasets with minimal overfitting”. As noted in the SI and main text, when reweighting a99SB-disp ensembles that are already in excellent agreement with experiment, of the fifty experimental data-sets used to reweight IDP ensembles – 47/50 data-sets withheld in leave-one-out cross-validation have an improved RMSD compared to unbiased MD – which is remarkable result given that experimental data collected on different samples can frequently be in tension. We believe that obtaining these striking cross-validation results with a single free parameter (σ_{reg}), without requiring multiple explicit additional terms for experimental error estimates, is a main strength of our proposed reweighting procedure.

(2) Role of regularization parameters?

One major point by the authors is that they have developed a procedure to estimate regularization parameters to weigh the different experimental data types, independent of the actual experimental errors. They do estimate the sampling errors in molecular simulations, σ_{MD} , though. I am wondering if estimating σ_{MD} is even necessary. Would the authors get the same results using their procedure if all σ_{MD} were set to zero?

If σ_{MD} are zero, then their regularization parameter σ_{reg} as well as their global rescaling factor $\sigma_{\text{reg-global}}$ would enter a multiplicative parameter for each data type. If so, this distinguishing property of their approach from previous approaches would not be needed. If this is not the case and their approach of using σ_{MD} explicitly has clear advantages, then this should be clarified and best illustrated with reweighting results.

Below we present an example of performing reweighting with and without σ_{MD} when reweighting the drkN SH3 a99SB-disp ensemble.

Example of effect of σ_{MD} in Proposed Reweighting Procedure

drkN SH3 a99SB-*disp* MD ensemble

Review Response Figure 1: Comparing reweighting of drkN sh3 a99SB-*disp* ensemble with and without σ_{MD} .

We observe that without σ_{MD} , the RMSE of some restrained data monotonically decrease as σ_{reg} approaches 0, indicating likely overfitting in this regime.

We note that in this example, we obtain better agreement with experimental RDCs at $Kish=0.10$ data when performing reweighting with σ_{MD} (RMSE=0.74 with σ_{MD} and RMSE = 0.80 with $\sigma_{MD}=0$), indicating that accounting for statistical noise of forward models with σ_{MD} predictions can facilitate obtaining a better balance between the weights of different restraints.

The reviewer is correct that in some instances, particularly where the initial ensemble is already in excellent agreement experiments, we find that including σ_{MD} does not have a substantial effect on the reweighting procedure or resulting ensemble. In these instances, however, we see no practical or conceptual disadvantage for including σ_{MD} .

We note the impact of σ_{MD} – which provides a measure of statistical noise of forward model predictions – could be especially important to avoid overfitting in future cases for data types not considered here, if one were to employ a noisy forward model with large statistical fluctuations. As such, we believe that including σ_{MD} in the reweighting protocol is a sensible precaution. We also reiterate that in our method, σ_{MD} is calculated for each experimental data point, and therefore enables regularization terms to be adopted for each data point based on the predictor noise for each prediction. Therefore, if a forward model produces particularly noisy predictions for a subset of data for a region of conformational space that is poorly described by a forward model, the regularization parameter for this data point will be adjusted accordingly.

Specifically, in the L-curve analysis mentioned by the authors, one uses a single multiplicative factor for the entropy. Decreasing this parameter decreases the Kish sampling size. In principle, one could pick the value of this factor that gives the desired Kish sampling size. Importantly, this parameter has the effect of a global multiplicative parameter for the total errors of the experimental data, similar to σ_{reg} -global if $\sigma_{MD}=0$. One would assume that using multiplicative scaling factors for individual data types, determined in a procedure similar to what the authors do, and using one global scaling factor will have a similar effect than their approach.

The reviewer is correct that it would be possible to implement a similar strategy to our proposed using L-curve analyses, and we have noted this in the “Comparisons to previous work” section of the SI.

Updated text (SI Page 22):

We note a conceptual similarity of the reweighting procedure proposed here with the concept of gentle ensemble refinement in the recently published work of Köfinger and Hummer. In this work, the authors propose an elegant approach to balance the strength of experimental restraints in maximum entropy reweighting against confidence in an initial prior model. They relate the KL divergence between the weights of an initial and reweighted ensemble to an energy uncertainty of the unbiased ensemble and propose to select the optimal value of restraint strength by relating the expected force field accuracy in the space of experimental observables to the expected energy variance upon reweighting. While motivated by theoretical considerations, this approach is similar in spirit and in practice to the heuristic approach proposed here, where we specify an acceptable limit to changes in the statistical weights of conformations in unbiased ensembles a priori. **We also note that one could implement a similar heuristic approach to one proposed here in the context of the L-curve/elbow method.**

(3) Best possible ensembles?

The authors chose to set a value for the Kish sampling size to compare force fields, which is fair. When it comes to the quality of the ensembles, fixing the sampling size for all force fields is expected to lead to sub-optimal ensembles for individual force fields. The Kish sampling size for the best force field might be smaller than necessary and (some of!) the experimental data might be overfitted, when taking into account experimental errors. For the worst force fields, agreement with the experimental data could perhaps be improved by lowering the Kish sampling size.

Although the authors provide good quality ensembles, I am wondering why not go the extra step and provide the best possible ensembles for each protein? After all, using one value of the Kish sampling size for all simulations to compare the ensembles puts a constraint on the quality of the reweighted ensemble.

As discussed in the SI section “Developing a maximum entropy reweighting protocol with a single free parameter” and illustrated in SI Figure 2, we emphasize that we consider selecting the “the best possible” ensemble to balance 1) the desired agreement with experimental data and 2) the desired level of statistical sampling in the reweighted ensembles.

In almost all instances considered in the manuscript, if one were to prioritize improved agreement with experimental data exclusively over the desired degree of statistical sampling in the reweighted ensembles, one could select a lower Kish threshold for reweighting – and this ultimately remains the prerogative of the user, depending on the desired application for the reweighted ensemble, and is a flexible feature in our code.

We have found however, that if one prioritizes only agreement with experimental data, the reweighted ensembles can become extremely sparse. In SI Figure 2, we demonstrate that as one increases the weights of restraints from $\sigma_{\text{reg}}=2.0$ to $\sigma_{\text{reg}}=1.0$, the effective ensemble size decreases from 3000 structures to ~100 structures, but the agreement with restrained data only negligibly improves, within the noise of the chemical shift predictors, likely as a result of overfitting.

For our research applications, we do not envision scenarios where highly sparse reweighted ensemble with 50-100 structures are more useful than an ensemble with 3000 structures with marginal worse agreement with experimental data. As the reviewer notes, this is of course always an option, and the Kish threshold is a free parameter that users can adjust to their liking.

We further demonstrate here the limited benefit (in our view) of substantially reducing the effective ensemble size in a scenario where all experimental data are used in restraints in **Review Response Figure 2**.

Review Response Figure 2: Comparing reweighting results of drkN sh3 a99SB-disp ensembles with small Kish Ratios.

Here we compare reweighting results obtained from an a99SB-disp simulation of drkN using a Kish ratio threshold of 0.10 (effective ensemble size of 3000 structures) and 0.002 (effective ensemble size of 60 structures). We note the RMSE of restrained data of the final ensembles (All Restraints) are relatively similar, and more importantly the differences in the RMSE of unrestrained cross-validation data are largely negligible, and in some instances (PRE-59, $^3\text{J}_{\text{HNH}\alpha}$) the agreement with cross-validating data is worse, suggesting overfitting.

For example, one could refine the combined ensembles for all force fields of a single protein into one and refine it against the experimental data. Their regularization approach should work. Force fields could have different initial probabilities, each of which has uniform initial weight.

While this is certainly a possibility, and something an interested user could decide to do with the code provided, we believe that maximum entropy reweighting is most meaningfully applied reweighting data from a single MD trajectory, as a tool to balance confidence in a given force field against a given set of experimental data. Merging MD ensembles from multiple ensembles inherently assumes that a user has equal confidence in the initial ensemble from each force field, which may not be a reasonable assumption when the initial ensembles have substantial differences in their agreement with experimental data. For example, in the cases of PaaA2 and a-synuclein, where the C22* and C36m ensembles are in substantially worse agreement with experiment than the a99SB-disp ensemble, merging the ensembles and performing reweighting with a Kish threshold=0.10 would result in a reweighted ensemble with worse agreement with experimental data than performing reweighting on the a99SB-disp ensemble alone. Nevertheless, this is certainly an option that users can explore with provided code.

(4) Central research question?

I do think that the central research question by the authors “with sufficient experimental data, can we determine physically realistic atomic-resolution IDP ensembles with conformational properties that are independent of the force fields used to generate the initial computational models?” can be easily tackled. I assume the authors agree.

For example, consistency between force fields could just mean that they do the same, not necessarily that they do it

correctly. Similarly, having agreement with experiment does not tell us how informative experimental data are when it comes to structural ensembles. Also, the issue of overfitting remains. In my opinion, the authors gave answers to the questions (1) How good are modern force fields when it comes to simulating IDPs? and (2) How can we reweight consistently when we want to compare force fields without using experimental errors? Although a step in the right direction, their results fall short of answering their central research question, in my opinion.

We acknowledge that the results of our study do not prove that the calculated ensembles are indeed true representations of the solution ensemble, and that, in particular, if one has a truly terrible initial MD ensemble, no reweighting procedure can produce a reweighted ensemble that provides a reasonable approximation of the true solution ensemble. **We have modified the text in several locations of the revised manuscript to reflect this important caveat.**

We note, however, that in these cases (as is the case in C22* and C36m ensembles of a-synuclein) – when applying our proposed reweighting procedure – which avoids extreme overfitting through the presence of the regularization parameters σ_{reg} and σ_{MD} – it is always clear when an ensemble cannot be meaningfully improved by reweighting. We believe a strength of the approach proposed here is that when given an initial MD ensemble in poor agreement with experiment, a property selected Kish threshold prevents overfitting to create the false impression of an accurate solution ensemble.

We have modified the abstract as follows (new text in bold):

Determining accurate atomic resolution conformational ensembles of intrinsically disordered proteins (IDPs) is extremely challenging. Molecular dynamics (MD) simulations provide atomistic conformational ensembles of IDPs, but their accuracy is highly dependent on the quality of physical models, or force fields, used. Here, we demonstrate how to determine accurate atomic resolution conformational ensembles of IDPs by integrating all-atom MD simulations with experimental data from nuclear magnetic resonance (NMR) spectroscopy and small-angle x-ray scattering (SAXS) with a simple, robust and fully automated maximum entropy reweighting procedure. **We demonstrate that in favorable cases, where IDP ensembles obtained from different MD force fields are in reasonable initial agreement with experimental data, reweighted ensembles obtained with this approach converge to highly similar conformational distributions.** The maximum entropy reweighting procedure presented here facilitates the integration of MD simulations with extensive experimental datasets and **demonstrates progress towards** the calculation of accurate, force-field independent conformational ensembles of IDPs at atomic resolution.

We modified the introduction as (new text in bold):

For three of the five IDPs studied, we find that the calculated conformational ensembles are highly similar, and can be considered a force-field independent approximation of the true underlying solution ensembles. In two of the IDPs studied, unbiased MD simulations performed with different force fields sample relatively distinct regions of conformational space, and our proposed reweighting method clearly identifies one ensemble as the most accurate representation of the true solution ensemble. **This demonstrates that in favorable cases, where IDP ensembles obtained from different MD force fields are in reasonable initial agreement with experimental data, reweighted ensembles obtained with the proposed approach converge to highly similar conformational distributions.**

We have modified the Results text on page 8 as follows (new text in bold):

We find that when we apply our proposed reweighting method to IDP ensembles obtained from long-timescale MD simulations with reasonably accurate force fields, the reweighted conformational ensembles are in excellent agreement with extensive experimental NMR and SAXS datasets that describe both local and global structural of IDPs. **We observe that for each protein studied here, we obtain at least one reweighted ensemble that is in substantially better agreement with experimental data than the most accurate unbiased MD ensembles of these proteins reported in previous benchmark studies.**

We have modified the Discussion on page (new text in bold):

We demonstrate, through extensive cross-validation, **that in favorable cases** the proposed reweighting approach simultaneously improves agreement with several types of experimental NMR data and SAXS data with minimal overfitting while maintaining a desired degree of sampling of the most populated regions of conformational space in unbiased MD ensembles. **We caveat however, that success of the proposed reweighting protocol is contingent on using a reasonably accurate, well-sampled initial ensemble. If an initial unbiased ensemble does not sample and appreciable population of conformational states that are consistent with experimental data, no reweighting protocol will be able to correct this distribution to resemble a realistic solution ensemble.**

We have also added an additional section in the SI “Importance of accuracy and adequate sampling in initial ensembles used for reweighting” to discuss this important issue in more detail:

Importance of accuracy and adequate sampling in initial ensembles used for reweighting

We emphasize that quality of reweighted ensembles obtained with the maximum entropy reweighting protocol proposed here, or obtained with any maximum entropy reweighting method, is highly dependent on the quality of the initial unbiased ensembles used for reweighting. As maximum entropy methods identify the minimal perturbation to an initial ensemble required to satisfy a given set of experimental restraints, the accuracy of reweighted ensembles is fundamentally limited by the compatibility of the structures in the initial ensemble with experimental data. For example, if an initial ensemble of an IDP is generated by a short MD simulation with an inaccurate force field that only samples overly collapsed states with a substantially smaller radius of gyration than the experimental radius of gyration, no reweighting method can produce an ensemble in agreement with the experimental SAXS data. Similarly, if experimental NMR chemical shifts indicate the presence of highly populated helical elements in an IDP ensemble, and an initial ensemble used for reweighting contains no helical conformations in these regions, it will not be possible to produce a reweighted ensemble with the correct populations of helical elements, regardless of strategies used to determine regularization parameters.

The success of any reweighting protocol is therefore contingent on using reasonably accurate, well-sampled initial ensembles as input for reweighting. IDP conformational ensembles are highly heterogeneous and contain structures with many distinct topologies separated by large free-energy barriers. Sampling this vast conformational in explicit solvent all-atom MD simulations is extremely challenging. Therefore, even if an accurate force field and water model combination is used for MD

simulations very long-timescale MD simulations (10s-100s of ms) or enhanced sampling methods are generally required to obtain statistically meaningful descriptions of IDP conformational ensembles.

Additional comments in arbitrary order

• I think readers could benefit from more guidance when it comes to the energy landscape visualization method (ELViM), used prominently by the authors. The energy landscape visualization method is not (yet?) an established method. To understand the plots, one has to have some minimal understanding of this method: In this method, each point in the 2D projection corresponds to a structure in the ensemble. Points are arranged such that the distance between points correlates with the dissimilarity of these two structures. Similar structures are close-by and form density peaks. Consequently, individual projections are not comparable. To overcome this problem, the authors combine the ensembles they want to compare in a single ensemble for which they then perform ELViM. They then separate the individual ensembles in these projections for analysis, e.g., for calculating the overlap integral. I do not know if their approach is original, already contained in the ELViM paper, or in the paper they have been inspired by (Reference 11 of the manuscript). Anyhow, I think this important point deserves clarification.

Indeed, when comparing ensembles using ELViM, it is necessary to project the two ensembles in the same latent space, and this is indeed noted in the original ELViM publication and emphasized in our manuscript.

We note that in the original ELViM paper, they do not quantify the overlap of the resulting ensembles as we do in this manuscript. However, as comparing the overlap of conformational space in different projections is a relatively common practice, with many different approaches found in the literature, we have chosen not to emphasize this feature as a notably novel portion of this manuscript.

• Are error bars available for SAXS data (e.g., for ACTR in Figure 2 A)? SAXS experiments on IDPs are difficult and SAXS intensities are often not that informative. Usually, the radius of gyration is the most important experimental information but often contaminated by aggregation. At higher angles, SAXS intensities of IDPs are usually quite featureless. Without error bars it is impossible to judge even visually the quality of the data and thus the quality of the fits. This question relates again to overfitting.

We thank the reviewer for raising this important point.

We have now added experimental SAXS error bars for ACTR and PaaA2 to Figure 2 and Figure 3 and a-synuclein in SI Figure 21. We include the modified figure panels showing agreement with experimental data before and after reweighting here:

ACTR (Figure 2)

PaaA2 (Figure 3)

a-synuclein (SI Figure 21)

• In Fig. 3 A, the SAXS data seems to have a significantly different radius of gyration than the reweighted ensemble for a99SB-disp. The fit is even worse in SI Fig. 12. Are calculated radii of gyration significantly different from the experimental data? Without error bars, it is hard to judge how to think about these fits.

Inspection of Guinier regions of SAXS data in Figure 2, Figure 3 and SI Figure 21 with added experimental errors shows substantial differences in the unbiased and reweighted ensembles of each force field. For additional clarification, we directly compare the unbiased and reweighted Rg values of each ensemble for a-synuclein, PaaA2, and drkN SH3 below to the experimental values determined by a Guinier analysis in **Review Response Figure 3**.

We note the differences in R_g are substantially larger than the experimental error estimates.

α -syn Rgs (exp $R_g=3.1 \pm 0.5$)			Paaa2 Rgs (exp $R_g=2.24 \pm 0.4$)		
System	Unbiased R_g	Reweighted R_g	System	Unbiased R_g	Reweighted R_g
a99SB- disp	3.56 ± 0.005	3.73 ± 0.01	a99SB- disp	2.08 ± 0.003	2.17 ± 0.006
Charmm36m	1.83 ± 0.001	1.98 ± 0.003	Charmm36m	1.37 ± 0.001	1.44 ± 0.002
c22star-TIP3P	1.62 ± 0.001	1.68 ± 0.002	c22star-TIP3P	1.33 ± 0.001	1.41 ± 0.003

drkN SH3 Rgs (exp $R_g=1.67 \pm 0.14$)

System	Unbiased R_g	Reweighted R_g
a99SB- disp	1.91 ± 0.002	1.80 ± 0.007
Charmm36m	1.51 ± 0.003	1.60 ± 0.006
c22star-TIP3P	1.39 ± 0.002	1.50 ± 0.005

Review Response Figure 3: Comparison of R_g values of unbiased and reweighted ensembles to experimental R_g measurements.

• I am wondering about the systematic errors in the calculations of the experimental observables (see, for example, Frohliking et al. JCP 158 2023). These calculations depend often on the choice of parameter values, which themselves are uncertain. This uncertainty implies systematic errors. I assume these parameter uncertainties can have dramatic effects on the quality of the reweighting.

Indeed, inaccurate forward models with large systematic errors will not be useful for reweighting. We believe the accuracy and uncertainties of the forward models used in this work are well described in their original publications, and in subsequent publications employing these models for reweighting.

In our proposed method, we view σ_{MD} and σ_{reg} as terms that are sufficiently large to also account for error terms that are not individually represented with their own separate regularization parameters (including random and systematic experimental errors as well as random and systematic errors in forward model predictors). We believe that excellent results of leave-one-out cross-validation results presented in our study demonstrate that we are not overfitting to predictor noise.

• For example, when the authors fit SAXS intensities, do they use a scaling parameter and a shift parameter for the intensities in the fitting? The shift parameter not only corrects for errors in the background subtraction but also accounts for contrast differences.

We follow the fitting procedure described by Pesce and Lindorff (Biophysical Journal, 2021 Nov 16;120(22):5124-5135. doi: 10.1016/j.bpj.2021.10.003.) which includes a scaling parameter and a shift parameter. We reference this work in our manuscript, and the shift and scaling parameters are implemented in our accompanying code.

• The authors mention that they follow the formalism of Bussi et al. If I am not mistaken, Bussi et al. have shown in the supporting information of Cesari et al. JCTC 2016 that for Gaussian noise, which the authors use here, their approach is equal to the approach of Hummer and Kofinger JCP 2015. If so, shouldn't this be clearly pointed out?

In the field of ensemble refinement, many approaches that superficially seem different actually are the same at the core. Being clear about it benefits the field and possible new users, in my opinion.

We believe that the similarity between the reweighting approaches of Cesari & Bussi and Hummer and Kofinger are thoroughly described in the original publication of Cesari et al. JCTC 2016, and have been similarly discussed in additionally discussed in later publications by Hummer and Kofinger and Bussi and co-workers.

We believe that we clearly note that our proposed method can be viewed as an extension or modification of the reweighting approach of Cesari and Bussi – and that we clearly describe the differences in our approaches in the methods section “*A maximum entropy reweighting protocol with a single free parameter*” and the SI section “*Overview of maximum entropy reweighting*”

• Do the authors have some data or rationale to corroborate that ensembles are well-sampled in simulations? Sampling must depend on chain length and might depend on secondary structure content, for example. The quality of sampling must be different for different IDPs. It would be great to get to know more about the quality of sampling, when it comes to different structural properties. Block averaging does not give all the information needed to judge how well some properties have been sampled.

We have added additional discussion of the importance of sampling in the discussion section and a new supporting information section “***Importance of accuracy and adequate sampling in initial ensembles used for reweighting***”. We now further emphasize the important caveat that a poorly sampled or inaccurate IDP ensemble is unlikely to meaningfully improve or provide a reasonable approximation of the true solution ensemble using any reweighting method.

We note that quantifying the sampling of IDP ensembles is a challenging open and active area of research (with no prescriptive or widely adopted solutions or metrics beyond computing statistical errors of simulated properties or free energy errors from different trajectory projections with methods such as a blocking) which is not the focus of the present manuscript.

However, we believe that by presenting comparisons of the similarity of the each unbiased and reweighted MD projected onto the ELViM latent space (Figures 4-6, SI Figures 25-27), the Rg vs. $S\alpha$ latent space (SI Figures 5,9,14,18,23), and now in the revised manuscript, projections onto an additional latent space obtained by performing PCA on interatomic distances (SI Figure 28), in addition to comparing the secondary structures and contact maps of each ensemble before and after reweighting (SI Figures 6,10,15,19,24) our manuscript provides readers with several informative orthogonal comparisons of regions of conformational space sampled in each trajectory.

We additionally highlight that these previously published trajectories are freely available enabling interested readers are able download the trajectories and assess convergence using any metric of their choosing. While the development of methods to quantify convergence of IDP simulations is of broad interest to the computational biophysics community, we do not believe that attempting to develop or apply additional convergence metrics will affect the conclusions of our study, where we limit our applications to 30us unbiased MD simulations, enabling fair comparisons between all force fields.

• I am not sure that the emphasis on “a single free parameter” gives the right impression. Other methods also have a single free parameter, but rely on estimates of the theoretical and experimental errors. If I understand the approach of the authors correctly, it is more about being able to refine without knowing experimental errors.

We believe a strength of our method is the use of a single free parameter, and the proposed method effectively solves an important practical problem of how to effectively combine different sources of experimental data without having to manually adjust the relative strengths of different restraints and demonstrate that the proposed approach produces minimal overfitting using leave-one-out cross validation analyses.

We note that accounting for experimental errors is just one portion of deciding how to adjust the relative weights of restraint and that accounting for experimental errors alone does not provide a prescriptive, deterministic method for balancing the weights of different classes of experimental restraints, which is an important feature of the reweighting method proposed in our work.

Reviewer #2 (Remarks to the Author):

In this paper, the authors integrated all-atom MD simulations with experimental data, namely NMR and SAXS using automated maximum entropy reweighting procedure. They applied this method to five well-studied intrinsically disordered proteins and obtained consistent structure ensembles from MD simulations based on different force-field parameters.

The methods are successful for three IDPs, while the results of the other two IDPs are not so impressive because of the separation of the unbiased MD ensembles with different force fields. These two are larger proteins, and enhanced conformational sampling in unbiased MD simulations is critical in the method. Although they used 30-microsecond MD simulations, which seem to be sufficiently long in the current standard of the conventional MD simulations, they are not enough for reliable reweighted structural ensembles.

Therefore, I think some of the sentences in this paper seem to be overstated.

The limitation highlights the importance of enhanced conformational sampling methods and/or multi-scale MD simulations to explore wide conformational spaces of intrinsically disordered proteins. Although the authors discussed enhanced sampling in the main text, they did not cite critical studies on developing enhanced conformational sampling methods. I suggest the authors add more citations to those works.

These statements are valid only when sufficient conformational space is explored in unbiased MD simulations. I strongly suggest the authors change these statements and add the limitations of these methods.

We thank the reviewer for raising several important points that were not sufficiently addressed in the in previous version of manuscript, in particular in regard to the fact that if one begins with an inaccurate ensemble with poor sampling, there is nothing that any reweighting method can do to produce a “force-field independent ensemble”. **We have added additional clarification in the abstract introduction and discussion to emphasize this important caveat.**

In Abstract, “We demonstrate that when this approach is applied with sufficient experimental data, IDP ensembles derived from different MD force fields converge to highly similar conformational distributions.”

We have modified the abstract as follows (new text in bold):

We demonstrate that in favorable cases, where IDP ensembles obtained from different MD force fields are in reasonable initial agreement with experimental data, reweighted ensembles obtained with this approach converge to highly similar conformational distributions. The maximum entropy reweighting procedure presented here facilitates the integration of MD simulations with extensive experimental datasets and **demonstrates progress towards** the calculation of accurate, force-field independent conformational ensembles of IDPs at atomic resolution.

In Results on page 8, “The conformational ensembles calculated in this work are in exceptionally good agreement with extensive experimental NMR and SAXS datasets, that describe both local and global structural properties of IDPs, and are in substantially better agreement with experimental data than the most accurate unbiased MD ensembles of these proteins reported in previous benchmark studies.”

We have modified the Results text on page 8 as follows (new text in bold):

We find that when we apply our proposed reweighting method to IDP ensembles obtained from long-timescale MD simulations with reasonably accurate force fields, the reweighted conformational ensembles are in excellent agreement with extensive experimental NMR and SAXS datasets that describe both local and global structural of IDPs. We observe that for each protein studied here, we obtain at least one reweighted ensemble that is in substantially better agreement with experimental data than the most accurate unbiased MD ensembles of these proteins reported in previous benchmark studies.

In Discussion on page 12, “We demonstrate, through extensive cross-validation, that the proposed reweighting approach simultaneously improves agreement with several types of experimental NMR data and SAXS data with minimal overfitting while maintaining a desired degree of sampling of the most populated regions of conformational space in unbiased MD ensembles.”

We have modified the Discussion text as follows (new text in bold):

...We demonstrate, through extensive cross-validation, **that in favorable cases** the proposed reweighting approach simultaneously improves agreement with several types of experimental NMR data and SAXS data with minimal overfitting while maintaining a desired degree of sampling of the most populated regions of conformational space in unbiased MD ensembles.

We caveat, however, that the success of the reweighting protocol proposed here, and all maximum-entropy reweighting protocols in general, is contingent on using a reasonably accurate, well-sampled initial ensemble as input for reweighting. If an initial unbiased ensemble does not sample an appreciable population of conformational states that are consistent with experimental data, no reweighting protocol will be able to correct this distribution to resemble a realistic solution ensemble.

We have added additional discussion of enhanced sampling in the SI section “*Importance of accuracy and adequate sampling in initial ensembles used for reweighting*”

Importance of accuracy and adequate sampling in initial ensembles used for reweighting

We emphasize that quality of reweighted ensembles obtained with the maximum entropy reweighting protocol proposed here, or obtained with any maximum entropy reweighting method, is highly dependent on the quality of the initial unbiased ensembles used for reweighting. As maximum entropy methods identify the minimal perturbation to an initial ensemble required to satisfy a given set of experimental restraints, the accuracy of reweighted ensembles is fundamentally limited by the compatibility of the structures in the initial ensemble with experimental data. For example, if an initial ensemble of an IDP is generated by a short MD simulation with an inaccurate force field that only samples overly collapsed states with a substantially smaller radius of gyration than the experimental radius of gyration, no reweighting method can produce an ensemble in agreement with the

experimental SAXS data. Similarly, if experimental NMR chemical shifts indicate the presence of highly populated helical elements in an IDP ensemble, and an initial ensemble used for reweighting contains no helical conformations in these regions, it will not be possible to produce a reweighted ensemble with the correct populations of helical elements, regardless of strategies used to determine regularization parameters.

The success of any reweighting protocol is therefore contingent on using reasonably accurate, well-sampled initial ensembles as input for reweighting. IDP conformational ensembles are highly heterogeneous and contain structures with many distinct topologies separated by large free-energy barriers. Sampling this vast conformational space in explicit solvent all-atom MD simulations is extremely challenging. Therefore, even if an accurate force field and water model combination is used for MD simulations, very long-timescale MD simulations (10s-100s of μ s) or enhanced sampling methods^(New Refs) are generally required to obtain statistically meaningful descriptions of IDP conformational ensembles.

And we have added the additional references:

New Refs:

(33) Appadurai, R.; Nagesh, J.; Srivastava, A. High resolution ensemble description of metamorphic and intrinsically disordered proteins using an efficient hybrid parallel tempering scheme. *Nature communications* 2021, 12, 958.

(34) Zhang, Y.; Liu, X.; Chen, J. Re-Balancing Replica Exchange with Solute Tempering for Sampling Dynamic Protein Conformations. *Journal of chemical theory and computation* 2023, 19, 1602–1614.

(35) Bussi, G.; Laio, A. Using metadynamics to explore complex free-energy landscapes. *Nature Reviews Physics* 2020, 2, 200–212.

(36) Gong, X.; Zhang, Y.; Chen, J. Advanced sampling methods for multiscale simulation of disordered proteins and dynamic interactions. *Biomolecules* 2021, 11, 1416.

The energy landscape visualization method (ELViM) is used for dimensionality reduction. It looks interesting; however, the authors did not compare the performance and advantages of ELViM with other dimensionality reduction methods, such as PCA, tICA, UMAP, etc. The authors need to add more discussion about the comparison and advantages of ELViM.

We thank the reviewer for raising this important point. To address this point, we have now performed dimensionality reduction of the unbiased and reweighted ensembles using the more traditional PCA approach with inter-residue distances and compared overlap of the ensembles in the PCA latent space in a new figure (**SI Figure 28**). We note that we observe the same trends in differences in ensemble overlap before and after reweighting with ELViM and PCA and added text to reflect this in the main text.

Supplementary Figure 28: Comparison of the overlap of inter-residue distance PCA projections of unbiased and reweighted MD ensembles. A) 2D PCA projections were calculated for A β 40, drkN SH3, ACTR, PaaA2 and α -synuclein. PCA projections were calculated for each protein using all conformations from unbiased a99SB-*disp*, C22* and C36m MD ensembles. Projections of unbiased a99SB-*disp*, C22* and C36m MD ensembles on the PCA latent space are shown in the top row. In the bottom row, each point is colored with an opacity proportional to its statistical weight after reweighting with all available experimental data. More transparent points corresponding to structures lower statistical weights. Histograms were estimated for each projection using 60 bins in each dimension. Data points belonging to histogram bins that contain at least one data point from each force field are colored dark gray. B) Overlap of PCA latent space kernel densities of unbiased and reweighted IDP ensembles. Values in the blue triangles reflect the PCA density overlap of unbiased MD ensembles derived from different force fields, values in red triangles reflect the PCA density overlap of reweighted ensembles derived from different force fields, and the diagonal elements reflect the PCA density overlap of the reweighted ensemble derived from each force field with its parent unbiased MD ensemble.

To quantify the advantage of using the non-linear ELViM method to reduce the dimensionality of IDP ensembles, we have computed the silhouette score of both ELViM and PCA projections to quantify how well the distances between conformations in the 2D projections differentiate ensembles generated by different force fields.

Comparison of low-dimensional projections obtained from ELViM and PCA

We utilize the silhouette score⁶² (Eq. 13 and Eq. 14) to compare 2D ELViM and PCA projections in terms of their ability to distinguish between conformational ensembles generated from the 3 force fields investigated in this study: a99SB-*disp*, C22* and C36m. The silhouette score quantifies the consistency of clustering method by measuring how well each cluster is distinguished from it’s nearest neighbors in a lower dimensional projection. For each protein, we perform ELViM and PCA on all three simulation datasets in aggregate such that they are projected onto the same latent space. We consider each "cluster" to consist of all conformations belonging to a given "parent" unbiased MD simulation used for reweighting, and compute the silhouette score accordingly. Consequently, in the following, 'cluster' and 'parent MD simulation ensemble' are synonymous. Distances are measured using the Euclidean distance between projected points. To compute the silhouette score, let $a(i)$ be the average intra-cluster distance for point i in simulation ensemble a , and $b(i)$ be the average inter-cluster distance for point i and all of the points in nearest neighbor simulation ensemble, b . The silhouette score for point i is defined as:

$$s(i) = \frac{b(i) - a(i)}{\max\{a(i), b(i)\}} \quad (13)$$

The overall silhouette score is the average over all points:

$$\text{silhouette score} = \frac{1}{n} \sum_{i=1}^n s(i) \quad (14)$$

We observe that on average, we obtain substantially large silhouette scores from ELViM projections than PCA Projections (SI Table 1).

Table SI 1: Comparison of ELViM and PCA Silhouette Scores.

	α -Synuclein	PaaA2	ACTR	drkN-SH3	A β -40
ELViM	0.362	0.412	0.119	0.255	0.060
PCA	0.363	0.107	0.063	0.183	-0.034

Comparison of Silhouette scores⁶² (Eq. 13) to quantify the quality of low-dimensional projections of the conformational space sampled in MD simulation analyzed in this work.

In Figure 3A, I cannot find lines for C22/TIP3P. Please check this figure carefully.*

We thank the reviewer for bringing this to our attention. In the previous version of figure, the predicted SAXS curve of C36m obscures the curve of C22*, we have increased the thickness of the C22* line to make it more visible.

Reviewer #2 (Remarks on code availability):

They provided the code used in this study appropriately.

Reviewer #3 (Remarks to the Author):

In the manuscript entitled “Determining accurate conformational ensembles of intrinsically disordered proteins at atomic resolution”, Kaushik Borthakur, et al., present a maximum entropy reweighting scheme to obtain conformational ensembles of intrinsically disordered proteins (IDPs) which match experimental observables such as SAXS or NMR data. The unbiased ensembles are obtained by long (30 μ s) atomistic molecular dynamics simulations with different force fields.

The method presented and its application to IDPs is timely and relevant. Although a conceptually similar method has been presented by Köfinger and Hummer (J. Chem. Phys. 160, 114111 (2024), doi: 10.1063/5.0189901) as also mentioned by the authors, the presented test cases are more complex and take more experimental data into account.

The manuscript is potentially suitable for publication in Nature Communications after major revisions. My points are given below:

1) Out of the five test cases discussed, three result in a substantial improvement of the conformational ensembles, while the conformational ensembles of the different force fields remain distinct for two test cases. This dependence on the performance of an atomistic force field is a strength (considering e.g. side chain orientations), but also a serious limitation of the method. This limitation should be discussed clearly in the manuscript.

We thank the reviewer for raising this issue. We have added additional clarification in the abstract, introduction, discussion, and SI to emphasize this important point in the revised manuscript.

We have modified the abstract as follows (new text in bold):

We demonstrate that in favorable cases, where IDP ensembles obtained from different MD force fields are in reasonable initial agreement with experimental data, reweighted ensembles obtained with this approach converge to highly similar conformational distributions. The maximum entropy reweighting procedure presented here facilitates the integration of MD simulations with extensive experimental datasets and **demonstrates progress towards** the calculation of accurate, force-field independent conformational ensembles of IDPs at atomic resolution.

We have modified the Results text on page 8 as follows (new text in bold):

We find that when we apply our proposed reweighting method to IDP ensembles obtained from long-timescale MD simulations with reasonably accurate force fields, the reweighted conformational ensembles are in excellent agreement with extensive experimental NMR and SAXS datasets that describe both local and global structural of IDPs. We observe that for each

protein studied here, we obtain at least one reweighted ensemble that is in substantially better agreement with experimental data than the most accurate unbiased MD ensembles of these proteins reported in previous benchmark studies.

We have modified the Discussion text as follows (new text in bold):

...We demonstrate, through extensive cross-validation, **that in favorable cases** the proposed reweighting approach simultaneously improves agreement with several types of experimental NMR data and SAXS data with minimal overfitting while maintaining a desired degree of sampling of the most populated regions of conformational space in unbiased MD ensembles.

We caveat, however, that the success of the reweighting protocol proposed here, and all maximum-entropy reweighting protocols in general, is contingent on using a reasonably accurate, well-sampled initial ensemble as input for reweighting. If an initial unbiased ensemble does not sample and appreciable populations of conformational states that are consistent with experimental data, no reweighting protocol will be able to correct this distribution to resemble a realistic solution ensemble.

We have added additional discussion of enhanced sampling in the new SI section “Importance of accuracy and adequate sampling in initial ensembles used for reweighting”

Importance of accuracy and adequate sampling in initial ensembles used for reweighting

We emphasize that quality of reweighted ensembles obtained with the maximum entropy reweighting protocol proposed here, or obtained with any maximum entropy reweighting method, is highly dependent on the quality of the initial unbiased ensembles used for reweighting. As maximum entropy methods identify the minimal perturbation to an initial ensemble required to satisfy a given set of experimental restraints, the accuracy of reweighted ensembles is fundamentally limited by the compatibility of the structures in the initial ensemble with experimental data. For example, if an initial ensemble of an IDP is generated by a short MD simulation with an inaccurate force field that only samples overly collapsed states with a substantially smaller radius of gyration than the experimental radius of gyration, no reweighting method can produce an ensemble in agreement with the experimental SAXS data. Similarly, if experimental NMR chemical shifts indicate the presence of highly populated helical elements in an IDP ensemble, and an initial ensemble used for reweighting contains no helical conformations in these regions, it will not be possible to produce a reweighted ensemble with the correct populations of helical elements, regardless of strategies used to determine regularization parameters.

The success of any reweighting protocol is therefore contingent on using reasonably accurate, well-sampled initial ensembles as input for reweighting. IDP conformational ensembles are highly heterogeneous and contain structures with many distinct topologies separated by large free-energy barriers. Sampling this vast conformational in explicit solvent all-atom MD simulations is extremely challenging. Therefore, even if an accurate force field and water model combination is used for MD simulations very long-timescale MD simulations (10s-100s of μ s) or enhanced sampling methods are generally required to obtain statistically meaningful descriptions of IDP conformational ensembles.

2) Two out of the five test cases (PaaA2 and α -synuclein) do not show good agreement of the conformational ensembles between the different force fields and thus seem not to converge. Can the authors provide more details on the biophysical properties of these two test cases compared to the ones where the method was applied successfully? Why do the atomistic force fields used fail to produce sufficient number of similar conformations, which could be emphasized during the reweighting? Can the authors give some guidelines on the biophysical properties of the IDPs which might be favorable for the employed force fields?

This results from the well-documented property of simulations performed with CHARMM force fields and the TIP3P water model to substantially underestimate the R_g of longer IDPs. We have added additional text to reflect this issue (SI Page 21 “Importance of accuracy and adequate sampling in initial ensembles used for reweighting”)

We note that the unbiased ensembles of PaaA2 and α -synuclein obtained from the C22* and C36m force fields sample almost no conformations with an R_g as large as the experimental R_g determined from SAXS,^{16,18,19} due to the propensity of these force fields, used in combination with the TIP3P water model, to substantially underestimate the R_g of IDPs longer than 60 residues.¹⁶ As a result, the reweighted ensembles of these proteins do not converge to similar conformational distributions, reflecting the inherent limitations of maximum-entropy reweighting methods discussed above.

3) The authors hypothesize that longer sampling of the MD simulations could improve the conformational ensembles. To get a better agreement for the two problematic test cases, could it be also helpful to have access to more and/or different experimental data?

As now more clearly explained in the section “Importance of accuracy and adequate sampling in initial ensembles used for reweighting”, because no conformations with realistic R_g s are sampled in these ensembles, no amount of orthogonal experimental data will be able correct this deficiency.

4) How does the method perform for (larger) intrinsically disordered regions on structured proteins? Adding more test cases in this direction would be helpful for the community and emphasize the broad applicability of the presented maximum entropy reweighting protocol.

The method we propose here will be broadly application to disordered regions in the context of folded proteins, applying the same caveats about the importance of obtaining reasonably accurate initial ensembles discussed above. In this instance of long disordered regions in the context of structured domains, it will be important that initial ensembles are not generated with a force field where the disordered region is overly compact or overly attracted to folded domains. Regrettably, we currently do not have access to well-sampled MD simulations of long disordered regions in the context of structured proteins to add to this manuscript to illustrate this point, and generating such simulations with multiple force fields would take several months or years of simulation time on substantial compute resources.

5) Figure 6A compares the energy landscape visualization method (ELViM) projections of the unbiased and reweighted MD ensembles. In case of the reweighted ensembles (bottom row), bins which at least one data point from each force field are colored dark grey. How does this analysis look for the unbiased ensembles? Please include this analysis and discuss the improvement due to reweighting.

We clarify that as we do not change the projections after reweighting, the number of bins that contain at least one data point for each force field does not change before and after reweighting. We have updated the text reflect this.

6) *The method itself is only introduced in the Supplementary Information (SI). However, it is also part of the results in my view, and an overview should be given to the reader in the main manuscript. In the current version of the manuscript (particularly in the first part of the Results section), many references to sections in the SI are given, so that the text does not read very well.*

Due to page and word count limitations – and also to emphasize what is novel in our approach – we have chosen to highlight how our method deviates from previously proposed approaches in the main text, while recounting all mathematical details of the method in the SI.

7) *The α -helical order parameter S_α is not labelled consistently, with or without subscript.*

We thank the reviewer for drawing out attention to this. We have corrected this inconsistency.

8) *More details for reference 50 should be provided.*

We have added additional details for this reference

Full Reference:

Kish, L.: Survey Sampling. John Wiley & Sons, Inc., New York, London 1965, IX + 643 S., 31 Abb., 56 Tab., Preis 83 s.

We thank the reviewers for their constructive feedback. We updated the text of our manuscript to reflect this feedback. We provide a detailed point by point response below and note revisions to the text of the manuscript below.

REVIEWER COMMENTS

Reviewer #1 (Remarks to the Author):

I thank the authors for their response and modification to the manuscript. Overall, I am content with their answers and changes. However, I find some of the responses partially superficial and I do not completely agree with everything that was said by them. I refrain from going into details.

Instead, reading their rebuttal and the concerns of the other reviewers, I would like to share my thoughts on the significance of their results. Unfortunately, I find neither the method nor the results convincingly significant.

The work presented here set-out to achieve two main goals:

1) Develop a method for the structural biology and computational biophysics communities to integrate computer simulations with diverse experimental datasets to calculate accurate and statistically meaningful conformational ensembles of intrinsically disordered proteins (IDPs) with minimal overfitting to inform biophysical, structural, and drug discovery investigations that does not require any subjective, manual, or expert tuning of hyperparameters to balance the weights of different classes of restraints.

We believe that the excellent agreement of our reweighted ensembles obtained from 15 long timescale MD simulations of IDPs with varying structural properties (Shown in Figures 1-3, SI Figures 11,12, 21) and extensive leave-one-out cross validation analyses, described in the SI section "A maximum entropy reweighting protocol with a single free parameter improves agreement with extensive experimental datasets with minimal overfitting" and shown in SI Figures 3, 7,11,12,16,20, and 21 demonstrates that we have accomplished this goal.

2) Apply this method on several IDP systems to determine if conformational ensembles of obtained from long timescale MD simulations with different state-of-the-art force fields reweighted with extensive experimental datasets from NMR spectroscopy and SAXS beginning to converge to similar conformational distributions.

We believe that by presenting extensive comparisons of the conformational properties of unbiased and reweighted IDP ensembles obtained using the method developed in this work, including comparisons of a variety of structural properties (Figures 1-3, SI Figures 4-6, SI

Figures 8-10, SI Figures 12-15, SI Figures 17-19, SI Figures 21-24), and comparisons of ensembles projected onto the energy landscape visualization method (ELViM) latent space (Figures 4-6, SI Figures 25-27) and on a more conventional linear latent space obtained from PCA on interatomic distances (SI Figure 29), demonstrates that we have accomplished this goal.

As such we believe that both the methods and results presented here will be of broad interest to structural biology, computational biophysics and intrinsically disordered protein communities.

From the details and the results, it is not clear to me if the progress of the method is truly substantial. For example, I think I understand the attractiveness of the Kish sample size. The authors fix the Kish sampling size in their procedure. They could also have fixed the related Kulback-Leibler divergence instead, as has been done previously. Both quantities quantify the difference between initial and reweighted ensemble. I cannot judge if the strengths of the Kish sampling sizes outweighs their disadvantages compared to the Kullback-Leibler divergence. Although only a detail, it emphasizes the recipe character of their method, in my opinion. Also, I fail to understand why using a "single free parameter" is a hallmark of their method given the methods in the literature. That their method is "simple" is in the eye of the beholder. A method being "robust" is important, but this is also not a unique property of their approach.

We believe that the approach presented in this work to automatically balance the weights of different classes of experimental restraints without any manual or iterative tuning of reweighting hyperparameters presented in this manuscript represent an important contribution to field of structural biology and maximum-entropy reweighting, and that the results from our extensive cross validation analyses demonstrate that it is highly effective (SI Figures 3, 7,11,12,16, 20, and 21)

We note that the reviewer suggests that alternative approaches may have been possible to achieve this goal, and we certainly agree with this assertion. While we have chosen to utilize Kish ratios (or Kish sampling size) as a central heuristic to determine the relative weights of different classes of experimental restraints in automated fashion - it is certainly possible that other alternative approaches for this component of our method could be employed to achieve similar results.

The reviewer specifically mentions an alternative strategy involving "fixing the Kullback-Leibler divergence", an approach that is described in the 2024 Journal of Chemical Physics publication "Encoding prior knowledge in ensemble refinement" by Köfinger and Hummer (The Journal of Chemical Physics. 2024 Mar 21;160(11)), which was published a few months before we

submitted our manuscript. Indeed, in our SI section “Comparison to Previous Work” we discuss the conceptual similarity of our approach with the previously published concept of “gentle ensemble refinement” from this publication - where the Kullback-Leibler divergence between the weights of an initial and reweighted ensemble are related to the energy uncertainty of the unbiased ensemble, and propose to select the optimal value of restraint strength by relating the expected force field accuracy in the space of experimental observables to the expected energy variance upon reweighting.

Fixing the Kullback-Leibler divergence may indeed represent a viable alternative approach for this one aspect of our reweighting protocol. However, we emphasize that in our manuscript, we make no claims that similarly effective, robust and automated maximum-entropy reweighting procedures to calculate all-atom ensembles of IDPs from extensive sets of experimental data without manually tuning hyperparameters could not be obtained through other approaches. We hope that our work inspires the development of alternative approaches to accomplish this task, and that the reweighted ensembles we have calculated will serve as valuable benchmarks for such approaches.

We have now updated the discussion section of the main text and the SI text in “Comparisons to previous work” to better contextualize the results of our study and emphasize possible alternative approaches to utilizing the Kish ratio to determine the relative weights of experimental restraints **(new text in bold)**:

Main text Discussion Section:

While we have chosen to use the Kish ratios of reweighted ensembles as a central heuristic to determine the relative weights of different classes of experimental restraints in this work, it is possible that the Kullback-Leibler (KL) divergence of the weights initial and reweighted ensembles could be employed in a similar fashion[32], and that a similar heuristic approach to one proposed here could be employed in the context of the L-curve/elbow method.[28,29]

SI text in “Comparisons to previous work”:

*We note a conceptual similarity of the reweighting procedure proposed here with the concept of gentle ensemble refinement in the recently published work of Köfinger and Hummer[9]. In this work, the authors propose an elegant approach to balance the strength of experimental restraints in maximum entropy reweighting against confidence in an initial prior model. They relate the KL divergence between the weights of an initial and reweighted ensemble to an energy uncertainty of the unbiased ensemble and propose to select the optimal value of restraint strength by relating the expected force field accuracy in the space of experimental observables to the expected energy variance upon reweighting. **In the initial publication[9], this method was applied to perform reweighting using synthetic 1D and 2D datasets and to refine an all-atom ensemble of a 5-residue poly-alanine peptide with NMR scalar couplings that are only sensitive to backbone dihedral angles[9]. While motivated by theoretical considerations, this approach is similar in spirit and in practice to the heuristic approach proposed here, where we specify an acceptable limit to changes in the statistical weights of conformations in unbiased ensembles a***

priori. While we have chosen to use the Kish ratios of reweighted ensembles as a heuristic to determine the relative weights of different classes of experimental restraints in this work, it is possible that the KL divergence of the weights initial and reweighted ensembles could be employed in a similar fashion. We also note that one could implement a similar heuristic approach to one proposed here in the context of the L-curve/elbow method.

We have also added the following text to the main text discussion section the SI section "Introducing different source of errors" to more explicitly note the relationship between the formalism applied in this our work and previous work from Köfinger and Hummer

"We note that when the formalism of Bussi et al[1–3] is applied with gaussian noise, as described here, this approach is equivalent to the approach described by Hummer and Köfinger[6]

[6] Hummer, G. and Köfinger, J., 2015. Bayesian ensemble refinement by replica simulations and reweighting. *The Journal of chemical physics*, 143(24).

We note that the publication "Encoding prior knowledge in ensemble refinement" limits the application of this approach to reweighting synthetic data from a 1D double-well potential energy surface, synthetic data of 2D von Mises polymer model and reweighting an all-atom MD simulation a 5-residue polyalanine peptide with NMR J-couplings that are sensitive to only backbone dihedral angles. To our knowledge, this approach has yet to be extended to study ensembles of IDPs with several classes of experimental data that report on orthogonal structural features or used to compare reweighted IDP ensembles obtained from different force fields. We note that should the gentle ensemble refinement approach, or related approaches, be extended to calculate all-atom conformational ensembles of IDPs from extensive experimental data sets containing several different classes of restraints – the unbiased and reweighted structural ensembles of α -synuclein, PaaA2, A β 40, drkN SH3 and ACTR produced in our work (and deposited in the protein ensemble database – deposition codes pending) will provide valuable benchmarks and comparisons for assessing the differences in the accuracy and structural properties of ensembles obtained with different reweighting approaches.

I am also not confident that the quality of the reweighted ensembles is a significant enough improvement.

We emphasize that for each protein studied here – we obtain at least one reweighted ensemble that is in substantially better agreement with experimental data than the most accurate unbiased MD ensembles reported in extensive benchmark studies of state-of-the-art force fields.

One could say that the authors prefer consistency between ensembles over quality of the ensembles, which is fine.

We respectfully disagree with this characterization of our work.

We emphasize that in this work, our goal was to develop a prescriptive, automated and broadly applicable reweighting procedure that can be used by structural biologists and computational biophysics researchers to integrate extensive experimental datasets that report on orthogonal structural properties of IDPs to calculate physically realistic, statistically robust structural ensembles of IDPs from reasonably accurate all-atom MD simulations or generative modeling approaches that does not require any subjective, manual or expert tuning of hyperparameters to balance the relative weights of different classes of restraints.

At no point did we make any methodological decisions based on a “preference for consistency between ensembles over the quality of ensembles”. On the contrary – one of the major motivations of developing a method that does not require (or enable) researchers to manually adjust hyperparameters for each class of experimental restraints to tune the properties of reweighted IDP ensembles to their subjective preferences, was to enable unbiased comparisons between ensembles generated by different force fields.

Having developed a prescriptive, automated approach that eliminates the manual adjustment of reweighting hyperparameters, we are then able to assess the consistency of reweighted ensembles obtained from different force fields in an unbiased and objective fashion.

However, their goal of having consistent ensembles for different force fields does not align with the goal of having the best possible ensembles.

One can interpret the notion of “best possible ensembles” to mean “ensembles with closest agreement with experimental data”, however, as discussed in the SI section “Developing a maximum entropy reweighting protocol with a single free parameter” and illustrated in SI Figure 2, we emphasize that we consider selecting the “the best possible” ensemble to balance 1) the desired agreement with experimental data and 2) the desired level of statistical sampling in the reweighted ensembles.

In almost all instances considered in the manuscript, if one were to prioritize improved agreement with experimental data exclusively over the desired degree of statistical sampling in the reweighted ensembles, one could select a lower Kish threshold for reweighting – and this ultimately remains the prerogative of the user, depending on the desired application for the reweighted ensemble, and is a flexible feature in our code. The method we developed enables a user to adjust the desired Kish Ratio to produce an ensemble with the closest possible agreement to experimental data, given a desired level of statistical sampling in the reweighted ensemble.

As such, we disagree that there is any tension between the method we have developed and the goal of obtaining the “best possible ensemble”.

We have found however, that if one prioritizes only agreement with experimental data, the reweighted ensembles can become extremely sparse. In SI Figure 2, we demonstrate that as one increases the weights of restraints from $\sigma_{\text{reg}} = 2.0$ to $\sigma_{\text{reg}} = 1.0$, the effective ensemble size decreases from 3000 structures to ~ 100 structures, but the agreement with restrained data only negligibly improves, within the noise of the chemical shift predictors, likely as a result of overfitting. For our research applications, we do not envision scenarios where highly sparse reweighted ensemble with 50-100 structures are more useful than an ensemble with 3000 structures with marginal worse agreement with experimental data. As the reviewer notes, this is of course always an option, and the Kish threshold is a free parameter that users can adjust to their liking. We further demonstrate here the limited benefit (in our view) of substantially reducing the effective ensemble size in a scenario where all experimental data are used in restraints in **Review Response 2 - Figure 1**.

Review Response Figure 2: Comparing reweighting results of drkN sh3 a99SB-disp ensembles with small Kish Ratios.

Here we compare reweighting results obtained from an a99SB-disp simulation of drkN using a Kish ratio threshold of 0.10 (effective ensemble size of 3000 structures) and 0.002 (effective ensemble size of 60 structures). We note the RMSE of restrained data of the final ensembles (All Restraints) are relatively similar, and more importantly the differences in the RMSE of unrestrained cross-validation data are largely negligible, and in some instances (PRE-59, $^3\text{J}_{\text{HNH}\alpha}$) the agreement with cross-validating data is worse, suggesting overfitting.

For example, with the limited information provided by the authors, we have to assume that the initial ensembles suffer from limited sampling.

We note that quantifying the sampling of IDP ensembles is a challenging open and active area of research (with no prescriptive or widely adopted solutions or metrics beyond computing statistical errors of simulated properties or free energy errors from different trajectory projections with methods such as a blocking) which is not the focus of the present manuscript.

However, we believe that by presenting comparisons of the similarity of the each unbiased and reweighted MD projected onto the ELViM latent space (Figures 4-6, SI Figures 25-27), the R_g vs. S_α latent space (SI Figures 5,9,14,18,23), and now in the revised manuscript, projections onto an additional latent space obtained by performing PCA on interatomic distances (SI Figure 28), in addition to comparing the secondary structures and contact maps of each ensemble before and after reweighting (SI Figures 6,10,15,19,24) our manuscript provides readers with several informative orthogonal comparisons of regions of conformational space sampled in each trajectory. We additionally highlight that these previously published trajectories are freely available enabling interested readers are able download the trajectories and assess convergence using any metric of their choosing. As we provide a relatively extensive set of analyses comparing the conformational space sampling in each ensemble, in addition to the raw trajectory coordinates of unbiased and reweighted ensembles- we respectively disagree that we are providing "limited information".

While the development of methods to quantify convergence of IDP simulations is of broad interest to the computational biophysics community, we do not believe that attempting to develop or apply additional convergence metrics will affect the conclusions of our study, where we limit our applications to 30us unbiased MD simulations, enabling fair comparisons between all force fields

As pointed out by Reviewer 3, sampling could be improved by biasing, e.g., using the experimental data themselves. Such enhanced ensembles should improve the quality of the reweighed ensemble substantially. As the authors have pointed out, such biased sampling is challenging. Without it, though, the reweighting is not as useful as it could be. The provided ensembles are still useful though, perhaps the best atomistic ensembles of IDPs available. However, I personally would only use them with a healthy dose of skepticism.

The reviewer is correct to note that one could potentially improve the sampling and accuracy of ensembles before reweighting by applying experimental data as restraints. We have added the following text to discussion section of the main text and the SI Section "Importance of accuracy and adequate sampling in initial ensembles used for reweighting" (new text in bold)

Discussion section of main text:

*We caveat, however, that the success of the reweighting protocol proposed here, and all maximum-entropy reweighting protocols in general, is contingent on using a reasonably accurate, well-sampled initial ensemble as input for reweighting²⁶. If an initial unbiased ensemble does not sample appreciable populations of conformational states that are consistent with experimental data, no reweighting protocol will be able to correct this distribution to resemble a realistic solution ensemble. **In cases where existing physical models cannot provide reasonable starting points for reweighting, an additional option for improving the statistical sampling and accuracy of IDP ensembles prior to reweighting would be to incorporate experimental data as replica-averaged restraints[30,40,41] to bias the initial ensemble prior to reweighting.***

SI Section “Importance of accuracy and adequate sampling in initial ensembles used for reweighting”

An additional option for improving the statistical sampling and accuracy of IDP ensembles prior to reweighting would be to incorporate experimental data as replica-averaged restraints[6,37,38] to bias the initial ensemble prior to reweighting. We note however, that when performing experimentally biased sampling using several types of experimental data as restraints, one is still confronted with the challenge of determining the relative weights of different classes of experimental restraints.

In summary, I do not see sufficient significance of either their method nor their reweighted ensembles. I simply could not be convinced by the manuscript and the rebuttal. From the method point of view, it appears to me that their method is more like a recipe than a fundamental development. As we know, it is difficult to convince others of the advantages of their personal recipe. From the biological results point of view, not knowing how good their reweighted ensembles actually are strongly limits their usefulness. As we are all aware, assessing the quality of ensembles is intrinsically hard. We only know ensemble could be better.

We emphasize that the motivation of this work was to develop a method for the structural biology and computational biophysics communities to integrate computer simulations with diverse experimental datasets to calculate accurate and statistically meaningful conformational ensembles of intrinsically disordered proteins (IDPs) with minimal overfitting to inform biophysical, structural, and drug discovery investigations that does not require any subjective, manual, or expert tuning of hyperparameters to balance the weights of different classes of restraints.

We believe that the quality of an ensemble can be quantitatively assessed by its agreement with experimental data (Shown in Figures 1-3, SI Figures 11,12, 21) , and the robustness of a reweighting method can be quantitatively assessed by systematic leave-one-out cross-validation analyses (SI Figures 3, 7,11,12,16,20, and 21). Based on the quality of these results, obtained for several IDP systems with different force fields, we believe the methods and results presented here will be of broad interest to structural biology, computational biophysics and intrinsically disordered protein communities.

The reviewer mentions “The provided ensembles are still useful though, perhaps the best atomistic ensembles of IDPs available”. As such, we believe reweighted ensembles presented here, which will be made publicly available in the protein ensemble database – may function as the closest thing the structural biology and intrinsically disordered protein community currently has to “ground-truth” examples for future reweighting methods and increasingly popular generative modeling approaches to predict conformational ensembles of IDPs.

Furthermore, because our approach is prescriptive and fully automated, we envision the possibility of integrating the reweighting and ensemble comparison methods developed in this work into disordered protein structural databases (such as the protein ensemble database) – to enable automated benchmarking of such methods.

I am aware that my response is not as positive as the authors must have hoped for. Please keep in mind that I do think that their work is important and interesting. I am just not convinced it is significant enough progress. Luckily, there are two other reviewers and an editor who might think differently. Good luck!

We again thank the reviewer for their feedback and for taking the time to review this manuscript.